# Deep learning from phylogenies to uncover the epidemiological dynamics of outbreaks

J. Voznica[1,2,3 ✉], A. Zhukova[1,4,5,6 ✉], V. Boskova[7], E. Saulnier[1], F. Lemoine [1,4], M. Moslonka-Lefebvre[1] & O. Gascuel [1,8 ✉]

Widely applicable, accurate and fast inference methods in phylodynamics are needed to fully profit from the richness of genetic data in uncovering the dynamics of epidemics. Standard methods, including maximum-likelihood and Bayesian approaches, generally rely on complex mathematical formulae and approximations, and do not scale with dataset size. We develop a likelihood-free, simulation-based approach, which combines deep learning with (1) a large set of summary statistics measured on phylogenies or (2) a complete and compact representation of trees, which avoids potential limitations of summary statistics and applies to any phylodynamics model. Our method enables both model selection and estimation of epidemiological parameters from very large phylogenies. We demonstrate its speed and accuracy on simulated data, where it performs better than the state-of-the-art methods. To illustrate its applicability, we assess the dynamics induced by superspreading individuals in an HIV dataset of men-having-sex-with-men in Zurich. Our tool PhyloDeep is available on github. com/evolbioinfo/phylodeep.

[1] Institut Pasteur, Université Paris Cité, Unité Bioinformatique Evolutive, Paris, France. [2] Université de Paris, Paris, France. [3] Institut de Biologie de l'École Normale Supérieure, Ecole Normale Supérieure, CNRS, INSERM, Université Paris Sciences et Lettres, Paris, France. [4] Institut Pasteur, Université Paris Cité, Bioinformatics and Biostatistics Hub, Paris, France. [5] Institut Pasteur, Université Paris Cité, Epidemiology and Modelling of Antibiotic Evasion, Paris, France. [6] Université Paris-Saclay, UVSQ, Inserm, CESP, Villejuif, France. [7] Center for Integrative Bioinformatics Vienna, Max Perutz Labs, University of Vienna and Medical University of Vienna, Vienna, Austria. [8] Institut de Systématique, Evolution, Biodiversité (UMR 7205 - CNRS, Muséum National d'Histoire Naturelle, SU, EPHE, UA), Paris, France. ✉email: voznica.jakub@gmail.com; anna.zhukova@pasteur.fr; olivier.gascuel@mnhn.fr

Pathogen phylodynamics is a field combining phylogenetics and epidemiology[1]. Viral or bacterial samples from patients are sequenced and used to infer a phylogeny, which describes the pathogen's spread among patients. The tips of such phylogenies represent sampled pathogens, and the internal nodes transmission events. Moreover, transmission events can be dated and thereby provide hints on transmission patterns. Such information is extracted by phylodynamic methods to estimate epidemiological and population dynamic parameters[2–4], assess the impact of population structure[2,5], and reveal the origins of epidemics[6].

Birth-death models[7] incorporate easily interpretable parameters common to standard infectious-disease epidemiology, such as basic reproduction number $R_0$, infectious period, etc. In contrast to the standard epidemiological models, the birth-death models can be applied to estimate parameters from phylogenetic trees[8]. In these models, births represent transmission events, while deaths represent removal events for example due to treatment or recovery. Upon a patient's removal, their pathogens can be sampled, producing tips in the tree.

Here we focus on three specific, well-established birth-death models (Fig. 1): birth-death model (BD)[8,9], birth-death model with exposed and infectious classes (BDEI)[5,10,11], and birth-death model with superspreading (BDSS)[5,12]. These models were deployed using BEAST2[12,13] to study the phylodynamics of such diverse pathogens as Ebola virus[10], Influenza virus[12], Human

Immunodeficiency Virus (HIV)[5], Zika[14] or SARS-CoV-2[15]. Using these models, we will demonstrate the reliability of our deep learning-based approach.

While a great effort has been invested in the development of new epidemiological models in phylodynamics, the field has been slowed down by the mathematical complexity inherent to these models. BD, the simplest model, has a closed-form solution for the likelihood formula of a tree for a given set of parameters[8,10], but more complex models (e.g., BDEI and BDSS) rely on a set of ordinary differential equations (ODEs) that cannot be solved analytically. To estimate parameter values through maximum-likelihood and Bayesian approaches, these ODEs must be approximated numerically for each tree node[5,10–12]. These calculations become difficult as the tree size increases, resulting in numerical instability and inaccuracy[12], as we will see below.

Inference issues with complex models are typically overcome by approximate Bayesian computation (ABC)[16,17]. ABC is a simulation-based technique relying on a rejection algorithm[18], where from a set of simulated phylogenies within a given prior (values assumed for the parameter values), those closest to the analysed phylogeny are retained and give the posterior distribution of the parameters. This scheme relies on the definition of a set of summary statistics aimed at representing a phylogeny and on a distance measure between trees. The ABC approach is thus sensitive to the choice of the summary statistics and distance function (e.g., Euclidean distance). To address this issue Saulnier et al.[19] developed a large set of summary statistics. In addition, they used a regression step to select the most relevant statistics and to correct for the discrepancy between the simulations retained in the rejection step and the analysed phylogeny. They observed that the sensitivity to the rejection parameters were greatly attenuated thanks to regression (see also Blum et al.[20]).

Our work is a continuation of regression-based ABC, and aims at overcoming its main limitations. Using the approximation power of currently available neural network architectures, we propose a likelihood-free method relying on deep learning from millions of trees of varying size simulated within a broad range of parameter values. By doing so, we bypass the rejection step, which is both time-consuming with large simulation sets, and sensitive to the choice of the distance function and summary statistics. To describe simulated trees and use them as input for the deep learner, we develop two tree representations: (1) a large set of summary statistics mostly based on Saulnier et al.[19], and (2) a complete and compact vectorial representation of phylogenies, including both the tree topology and branch lengths. The summary statistics are derived from our understanding and knowledge of the epidemiological processes. However, they can be incomplete and thus miss some important aspects of the studied phylogenies, which can potentially result in low accuracy during inference. Moreover, it is expected that new phylodynamic models will require design of new summary statistics, as confirmed by our results with BDSS. In contrast, our vectorial representation is a raw data representation that preserves all information contained in the phylogeny and thus should be accurate and deployable on any new model, provided the model parameters are identifiable. Our vectorial representation naturally fits with deep learning methods, especially the convolutional architectures, which have already proven their ability to extract relevant features from raw representations, for example in image analysis[21,22] or weather prediction[23].

In the following, we introduce our vectorial tree representation and the new summary statistics designed for BDSS. We then present the deep learning architectures trained on these representations and evaluate their accuracy on simulated datasets in terms of both parameter estimation and model selection. We show that our approach applies not only to trees of the same size

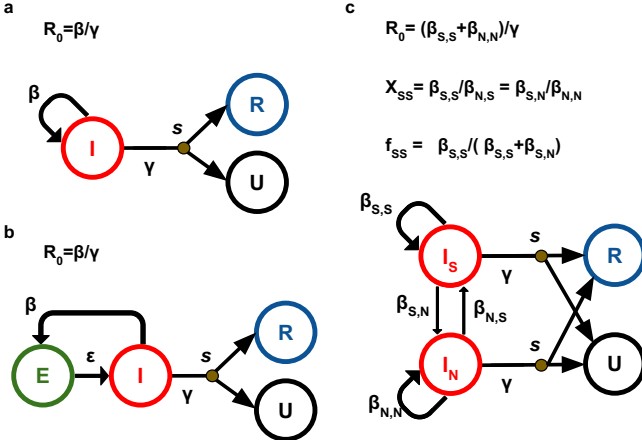

**Fig. 1 Birth-death models. a** Birth-death model (BD)[8,9], **b** birth-death model with Exposed-Infectious individuals (BDEI)[5,10,11] and **c** birth-death model with SuperSpreading (BDSS)[5,12]. BD is the simplest generative model, used to estimate $R_O$ and the infectious period $(1/\gamma)$[8,9]. BDEI and BDSS are extended version of BD. BDEI enables to estimate latency period $(1/\varepsilon)$ during which individuals of exposed class E are infected, but not infectious[5,10,11]. BDSS includes two populations with heterogeneous infectiousness: the so-called superspreading individuals (S) and normal spreaders (N). Superspreading individuals are present only at a low fraction in the population $(f_{ss})$ and may transmit the disease at a rate that is multiple times higher than that of normal spreaders (rate ratio = $X_{ss}$)[5,12]. Superspreading can have various complex causes, such as the heterogeneity of immune response, disease progression, co-infection with other diseases, social contact patterns or risk behaviour, etc. Infectious individuals I (superspreading infectious individuals $I_S$ and normal spreaders $I_N$ for BDSS), transmit the disease at rate $\beta$ ($\beta_{X,Y}$ for an individual of type X transmitting to an individual of type Y for BDSS), giving rise to a newly infected individual. The newly infected individual is either infectious right away in BD and BDSS or goes through an exposed state before becoming infectious at rate $\varepsilon$ in BDEI. Infectious individuals are removed at rate $\gamma$. Upon removal, they can be sampled with probability $s$, becoming of removed sampled class R. If not sampled upon removal, they move to non-infectious unsampled class U.

as the training instances, but also to very large trees with thousands of tips through the analysis of their subtrees. The results are compared to those of the gold standard method, BEAST2[12,13]. Lastly, we showcase our methods on an HIV dataset[24,25] from the men-having-sex-with-men (MSM) community from Zurich. All technical details are provided in 'Methods' and Supplementary Information. Our methods and tools are implemented in the PhyloDeep software, which is available on GitHub (github.com/evolbioinfo/phylodeep), PyPi (pypi.org/project/phylodeep) and Docker Hub (hub.docker.com/r/evolbioinfo/phylodeep).

## Results

Neural networks are trained on numerical vectors from which they can learn regression and classification tasks. We trained such networks on phylogenetic trees to estimate epidemiological parameters (regression) and select phylodynamic models (classification). We undertook two strategies for representing phylogenetic trees as numerical vectors, which we describe first, before showing the results with simulated and real data.

**Summary statistics (SS) representation**. We used a set of 83 SS developed by Saulnier et al.[19]: 26 measures of branch lengths, such as median of both internal and tip branch lengths; 8 measures of tree topology, such as tree imbalance; 9 measures on the number of lineages through time, such as time and height of its maximum; and 40 coordinates representing the lineage-through-time (LTT) plot. To capture more information on the phylogenies generated by the BDSS model, we further enriched these SS with 14 new statistics on transmission chains describing the distribution of the duration between consecutive transmissions (internal tree nodes). Our SS are diverse, complementary and somewhat redundant. We used feed-forward neural networks (FFNN) with several hidden layers (Fig. 2b (i)) that select and combine relevant information from the input features. In addition to SS, we provide both the tree size (i.e., number of tips) and the sampling probability used to generate the tree, as input to our FFNN (Fig. 2a (vi)). We will refer to this method as FFNN-SS.

**Compact vectorial tree representation**. While converting raw information in the form of a phylogenetic tree into a set of SS, information loss is unavoidable. This means not only that the tree cannot be fully reconstructed from its SS, but also that depending on how much useful and relevant information is contained in the SS, the neural network may fail to solve the problem at hand. As an alternative strategy to SS, and to prevent information loss in the tree representation, we developed a representation called 'Compact Bijective Ladderized Vector' (CBLV).

Several vectorial representations of trees based either on polynomial[26,27], Laplacian spectrum[28] or F matrices[29] have been developed previously. However, they represent the tree shape but not the branch lengths[26] or may lose information on trees[28]. In addition, some of these representations require vectors or matrices of quadratic size with respect to the number of tips[29], or are based on complex coordinate systems of exponential size[27].

Inspired by these approaches, we designed our concise, easily computable, compact, and bijective (i.e. 1-to-1) tree representation that applies to trees of variable size and is appropriate as machine learning input. To obtain this representation, we first ladderize the tree, that is, for each internal node, the descending subtree containing the most recently sampled tip is rotated to the left, Fig. 2a (ii). This ladderization step does not change the tree but facilitates learning by standardizing the input data. Moreover, it is consistent with trees observed in real epidemiological datasets, for example Influenza, where ladder-like trees reflect selection and are observed for several pathogens[1]. Then, we perform an inorder traversal[30] of the ladderized tree, during which we collect in a vector for each visited internal node its distance to the root and for each tip its distance to the previously visited internal node. In particular, the first vector entry corresponds to the tree height. This transformation of a tree into a vector is bijective, in the sense that we can unambiguously reconstruct any given tree from its vector representation (Supplementary Fig. 1). The vector is as compact as possible, and its size grows linearly with the number of tips. We complete this vector with zeros to reach the representation length of the largest tree contained in our simulation set, and we add the sampling probability used to generate the tree (or an estimate of it when analysing real data; Fig. 2a (v), b (i)).

Bijectivity combined with ladderization facilitates the training of neural networks, which do not need to learn that different representations correspond to the same tree. However, unlike our SS, this full representation does not have any high-level features. In CBLV identical subtrees will have the same representation in the vector whenever the roots of these subtrees have the same height, while the vector representation of the tips in such subtrees will be the same no matter the height of the subtree's root. Similar subtrees will thus result in repeated patterns along the representation vector. We opted for convolutional neural networks (CNN), which are designed to extract information on patterns in raw data. Our CNN architecture (Fig. 2b (ii)) includes several convolutional layers that perform feature extraction, as well as maximum and average pooling layers that select relevant features and keep feature maps of reasonable dimensions. The output of the CNN is then fed into a FFNN that combines the patterns found in the input to perform predictions. In the rest of the manuscript, we refer to this method as CNN-CBLV.

**Simulated datasets**. For each phylodynamic model (BD, BDEI, BDSS), we simulated 4 million trees, covering a large range of values for each parameter of epidemiological interest ($R_0$, infectious period: $1/\gamma$, incubation period: $1/\varepsilon$, the fraction at equilibrium of superspreading individuals: $f_{SS}$, and the superspreading transmission ratio: $X_{SS}$). Of the 4 million trees, 3.99 million were used as a training set, and 10,000 as a validation set for early stopping in the training phase[31]. In addition, we simulated another 10,000 trees, which we used as a testing set, out of which 100 were also evaluated with the gold standard methods, BEAST2 and TreePar, which are more time-consuming. Another 1 million trees were used to define confidence intervals for estimated parameters. For BD and BDEI we considered two settings: one with small trees (50 to 199 tips, in Supplementary Fig. 2) and a second with large trees (200 to 500 tips, Fig. 3). For BDSS, we considered only the setting with large trees, as the superspreading individuals are at a low fraction and cannot be detected in small trees. Lastly, we investigated the applicability of our approach to very large datasets, which are increasingly common with viral pathogens. To this goal, we generated for each model 10,000 'huge' trees, with 5000 to 10,000 tips each and with the same parameter ranges as used with the small and large trees. To estimate the parameter values of a huge tree, we extracted a nearly complete coverage of this tree by disjoint subtrees with 50 to 500 leaves. Then, we predicted the parameter values for every subtree using our NNs, and averaged subtree predictions to obtain parameter estimates for the huge tree.

To increase the generality of our approach and avoid the arbitrary choice of the time scale (one unit can be a day, a week, or a year), we rescaled all trees and corresponding epidemiological parameters, such that the average branch length in a tree was equal to 1. After inference, we rescaled the estimated parameter values back to the original time scale.

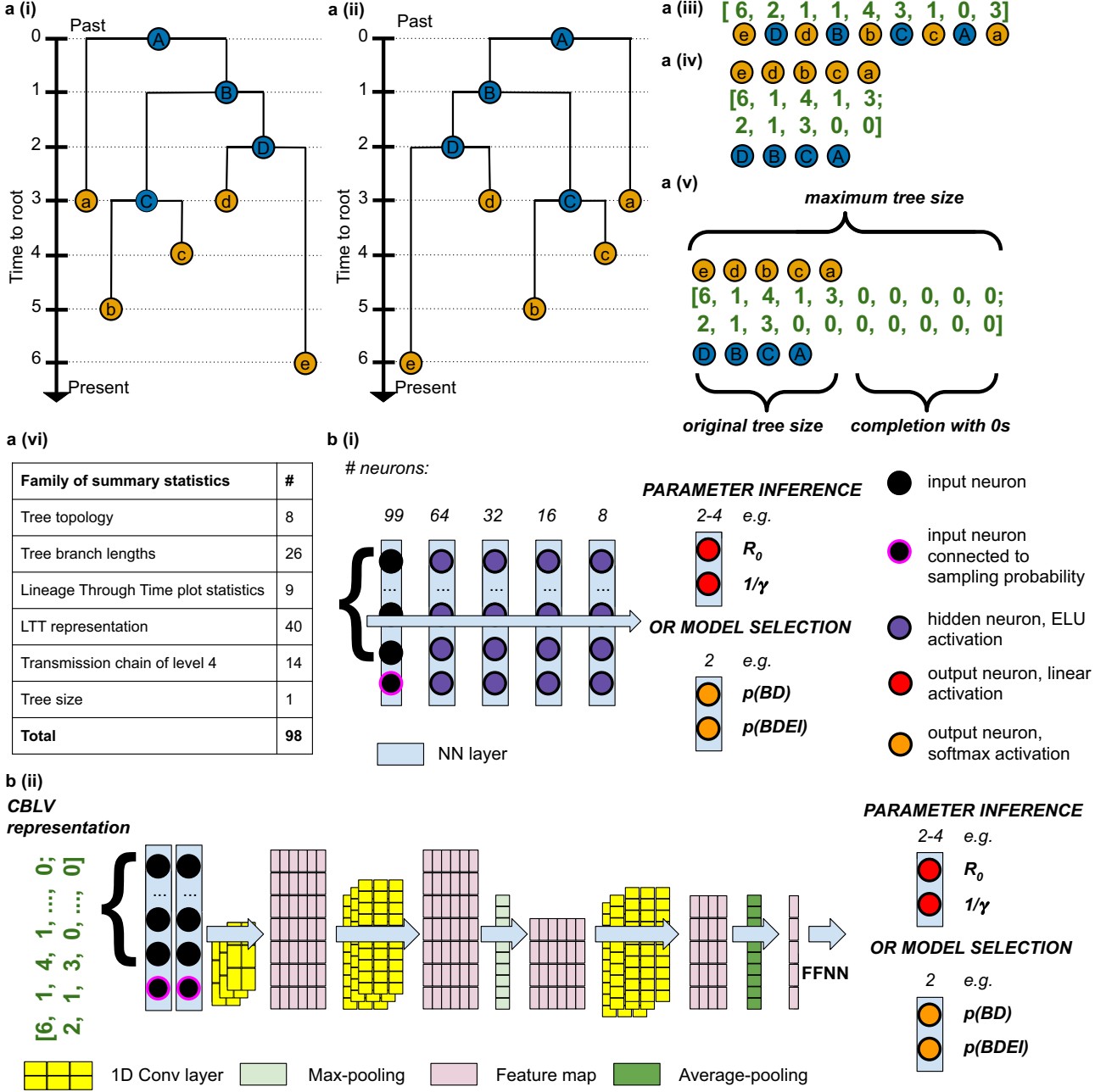

**Fig. 2 Pipeline for training neural networks on phylogenies.** Tree representations: **a** (i), simulated binary trees. Under each model from Fig. 1, we simulate many trees of variable size (50 to 200 tips for 'small trees' and 200 to 500 tips for 'large trees'). For illustration, we have here a tree with 5 tips. We encode the simulations into two representations, either **a** (ii–v), in a complete and compact tree representation called 'Compact Bijective Ladderized Vector' abbreviated as CBLV or **a** (vi) with summary statistics (SS). CBLV is obtained through **a** (ii) ladderization or sorting of internal nodes so that the branch supporting the most recent leaf is always on the left and **a** (iii) an inorder tree traversal, during which we append to a real-valued vector for each visited internal node its distance to the root and for each visited tip its distance to the previously visited internal node. We reshape this representation into **a** (iv), an input matrix in which the information on internal nodes and leaves is separated into two rows. Finally, **a** (v), we complete this matrix with zeros so that the matrices for all simulations have the size of largest simulation matrices. For illustration purpose, we here consider that the maximum tree size covered by simulations is 10, and the representation is thus completed with 0 s accordingly. SS consists of **a** (vi), a set of 98 statistics: 83 published in Saulnier et al.[19], 14 on transmission chains and 1 on tree size. The information on sampling probability is added to both representations. **b** Neural networks are trained on these representations to estimate parameter values or to select the underlying model. For SS, we use, **b** (i), a deep feed-forward neural network (FFNN) of funnel shape (we show the number of neurons above each layer). For the CBLV representation we train, **b** (ii), convolutional neural networks (CNN). The CNN is added on top of the FFNN. The CNN combines convolutional, maximum pooling and global average pooling layers, as described in detail in 'Methods' and Supplementary Information.

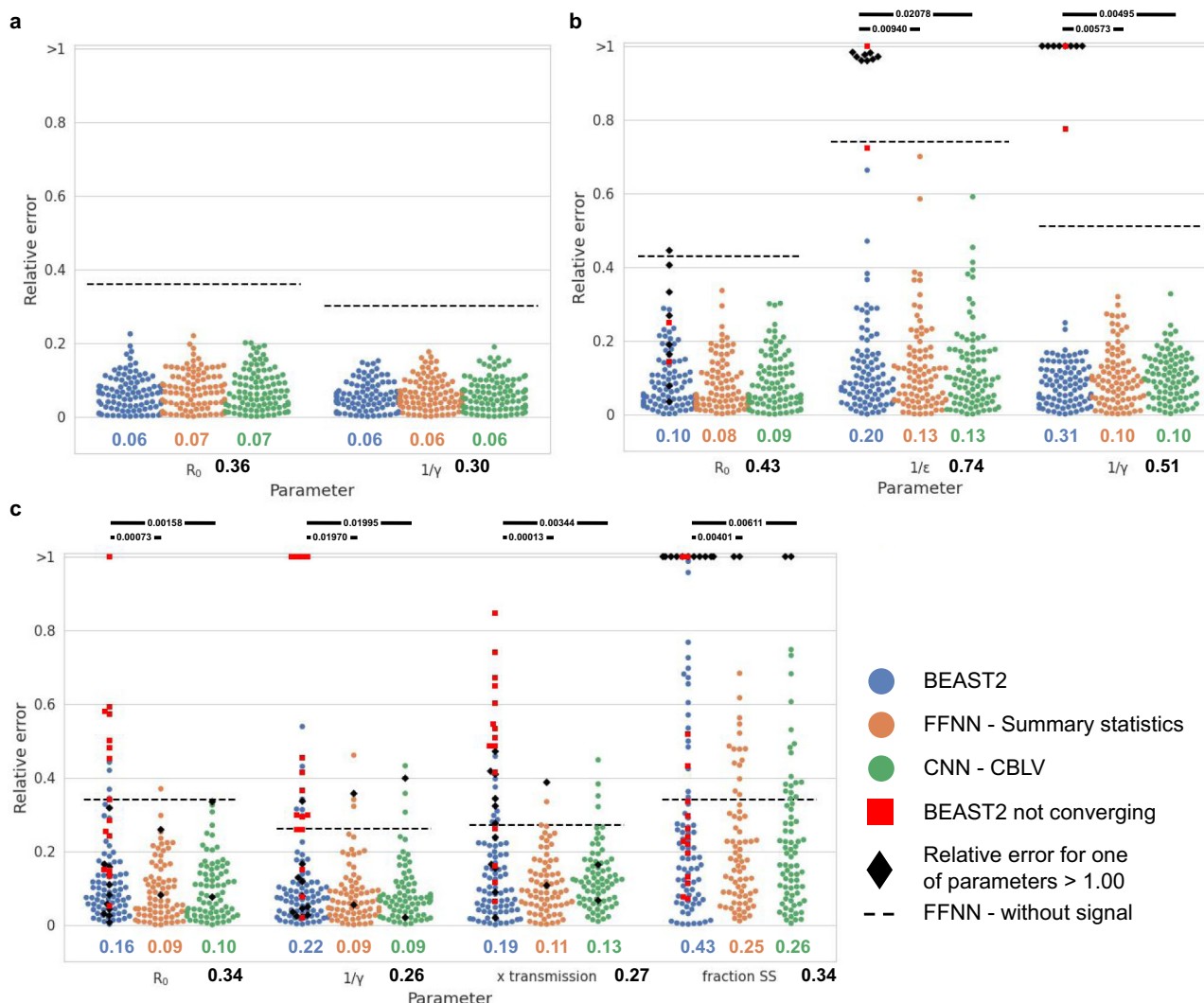

**Fig. 3 Assessment of deep learning accuracy.** Comparison of inference accuracy by BEAST2 (in blue), deep neural network trained on SS (in orange) and convolutional neural network trained on the CBLV representation (in green) on 100 test trees. The size of training and testing trees was uniformly sampled between 200 and 500 tips. We show the relative error for each test tree. The error is measured as the normalized distance between the median a posteriori estimate by BEAST2 or point estimates by neural networks and the target value for each parameter. We highlight simulations for which BEAST2 did not converge and whose values were thus set to median of the parameter subspace used for simulations, by depicting them as red squares. We further highlight the analyses with a high relative error (>1.00) for one of the estimates, as black diamonds. We compare the relative errors for **a** BD-simulated, **b** BDEI-simulated and **c** BDSS-simulated trees. Average relative error is displayed for each parameter and method in corresponding colour below each figure. The average error of a FFNN trained on summary statistics but with randomly permuted target is displayed as black dashed line and its value is shown in bold black below the x-axis. The accuracy of each method is compared by two-sided paired z-test; $P < 0.05$ is shown as thick full line; non-significant is not shown.

**Neural networks yield more accurate parameter estimates than gold standard methods.** We compared accuracy of parameter estimates yielded by our deep learning methods and those yielded by two state-of-the-art phylodynamics inference tools, BEAST2[12,13] and TreePar[5]. The comparison shows that our deep learning methods trained with SS and CBLV are either comparable (BD) or more accurate (BDEI and BDSS) than the state-of-the-art inference methods (Fig. 3, Supplementary Table 1). The simple BD model has a closed-form solution for the likelihood function, and thus BEAST2 results are optimal in theory[8,9]. Our results with BD are similar to those obtained with BEAST2, and thus nearly optimal as well. For BDEI and BDSS our results are more accurate than BEAST2, which is likely explained by numerical approximations of likelihood calculations in BEAST2[5,10,11] for these models. These approximations can lead BEAST2 to a lack of convergence (2% cases for BDEI and 15%

cases for BDSS) or a convergence to local optima. We suspect BEAST2 of converging to local optima when it converged to values with high relative error (>1.0; 8% cases for BDEI and 11% cases for BDSS, Fig. 3b, c). Furthermore, our deep learning approaches showed a lower bias in parameter estimation than BEAST2 (Supplementary Table 2). As expected, both approaches, FFNN-SS and CNN-CBLV, get more accurate with larger trees (Supplementary Fig. 3).

We tried to perform maximum-likelihood estimation (MLE) implemented in the TreePar package[5] on the same trees as well. While MLE under BD model on simulations yielded as accurate results as BEAST2, for more complex models it showed overflow and underflow issues (i.e., reaching infinite values of likelihood) and yielded inaccurate results, such as more complex models (BDEI, BDSS) having lower likelihood than a simpler, nested one (BD) for a part of simulations. These issues were more prominent

for larger trees. TreePar developers confirmed these limitations and suggested using the latest version of BEAST2 instead.

To further explain the performance of our NNs, we computed the likelihood value of their parameter estimates. This was easy with the BD model since we have a closed-form solution for the likelihood function. The results with this model (Supplementary Table 3, using TreePar) showed that the likelihoods of both FFNN-SS and CNN-CBLV estimates are similar to BEAST2's, which explains the similar accuracy of the three methods (Fig. 3). We also computed the likelihood of the 'true' parameter values used to simulate the trees, in order to have an independent and solid assessment. If a given method tends to produce higher likelihood than that of the true parameter values, then it performs well in terms of likelihood optimization, as optimizing further should not result in higher accuracy. The results (Supplementary Table 3) were again quite positive, as BEAST2 and our NNs achieved a higher likelihood than the true parameter values for ~70% of the trees, with a significant mean difference. With BDEI and BDSS, applying the same approach proved difficult due to convergence and numerical issues, with both BEAST2 and TreePar (see above). For the partial results we obtained, the overall pattern seems to be similar to that with BD: the NNs obtain highly likely solutions, with similar likelihood as BEAST2's (when it converges and produces reasonable estimates), and significantly higher likelihood than that of the true parameter values. All these results are remarkable, as the NNs do not explicitly optimize the likelihood function associated to the models, but use a radically different learning approach, based on simulation.

**Neural networks are fast inference methods**. We compared the computing time required by each of our inference methods. All computing times were estimated for a single thread of our cluster, except for the training of neural architectures where we used our GPU farm. Neural networks require heavy computing time in the learning phase; for example, with BDSS (the most complex model), simulating 4 M large trees requires ~800 CPU hours, while training FFNN-SS and CNN-CBLV requires ~5 and ~150 h, respectively. However, with NNs, inference is almost instantaneous and takes ~0.2 CPU seconds per tree on average, including encoding the tree in SS or CBLV, which is the longest part. For comparison, BEAST2 inference under the BD model with 5 million MCMC steps takes on average ~0.2 CPU hours per tree, while inference under BDEI and BDSS with 10 million MCMC steps takes ~55 CPU hours and ~80 CPU hours per tree, respectively. In fact, the convergence time of BEAST2 is usually faster (~6 CPU hours with BDEI and BDSS), but can be very long in some cases, to the point that convergence is not observed after 10 million steps (see above).

**Neural networks have high generalization capabilities and apply to very large datasets**. In statistical learning theory[31], generalization relates to the ability to predict new samples drawn from the same distribution as the training instances. Generalization is opposed to rote learning and overfitting, where the learned classifier or regressor predicts the training instances accurately, but new instances extracted from the same distribution or population poorly. The generalization capabilities of our NNs were demonstrated, as we used independent testing sets in all our experiments (Fig. 3). However, we expect poor results with trees that depart from the training distribution, for example showing very high $R_0$, while our NNs have been trained with $R_0$ in the range[1,5]. If, for a new study, larger or different parameter ranges are required, we must retrain the NNs with ad hoc simulated trees. However, a strength of NNs is that thanks to

their flexibility and approximation power, very large parameter ranges can be envisaged, to avoid repeating training sessions too often.

Another sensible issue is that of the size of the trees. Our NNs have been trained with trees of 50-to-199 tips (small) and 200-to-500 tips (large), that is, trees of moderate size (but already highly time-consuming in a Bayesian setting, for the largest ones). Thus, we tested the ability to predict the parameters of small trees using NNs trained on large trees, and vice versa, the ability to predict large trees with NNs trained on small trees. The results (Supplementary Fig. 4) are surprisingly good, especially with summary statistics (FFNN-SS) which are little impacted by these changes of scale as they largely rely on means (e.g., of branch lengths[19]). This shows unexpected generalization capabilities of the approach regarding tree size. Most importantly, the approach can accurately predict huge trees (Fig. 4) using their subtrees and the means of the corresponding parameter estimates, in ~1 CPU minute. This extends the applicability of the approach to datasets that cannot be analysed today, unless using similar tree decomposition and very long calculations to analyse all subtrees.

**Neural networks are accurate methods for model selection**. We trained CNN-CBLV and FFNN-SS on simulated trees to predict the birth-death model under which they were simulated (BD or BDEI for small trees; BD, BDEI or BDSS for large trees). Note that for parameters shared between multiple models, we used identical parameter value ranges across all these models (Supplementary Table 4). Then, we assessed the accuracy of both of our approaches on 100 simulations obtained with each model and compared it with the model selection under BEAST2 based on Akaike information criterion through Markov Chain Monte Carlo (AICM)[32,33]. The AICM, similar to deviance information criterion (DIC) by Gelman et al.[32], does not add computational load and is based on the average and variance of posterior log-likelihoods along the Markov Chain Monte Carlo (MCMC).

FFNN-SS and CNN-CBLV have similar accuracy (Supplementary Table 5), namely 92% for large trees (BD vs BDEI vs BDSS), and accuracy of 91% and 90%, respectively, for small trees (BD vs BDEI). BEAST2 yielded an accuracy of 91% for large trees and 87% for small trees. The non-converging simulations were not considered for any of these methods (i.e., 5% simulations for small trees and 24% for large trees).

The process of model selection with a neural network is as fast as the parameter inference (~0.2 CPU seconds per tree). This represents a practical, fast and accurate way to perform model selection in phylodynamics.

**Neural networks are well suited to learn complex models**. To assess the complexity of learned models, we explored other inference methods, namely: (1) linear regression as a baseline model trained on summary statistics (LR-SS); (2) FFNN trained directly on CBLV (FFNN-CBLV); (3) CNN trained on Compact Random Vector (CNN-CRV), for which the trees were randomly rotated, instead of being ladderized as in Fig. 2 (ii); and (4) two "null models".

LR-SS yielded inaccurate results even for the BD model (Supplementary Table 1), which seems to contrast with previous findings[19], where LR approach combined with ABC performed only slightly worse than BEAST2. This can be explained by the lack of rejection step in LR-SS, which enables to locally reduce the complexity of the relation between the representation and the inferred values to a linear one[18]. However, the rejection step requires a metric (e.g., the Euclidean distance), which may or may

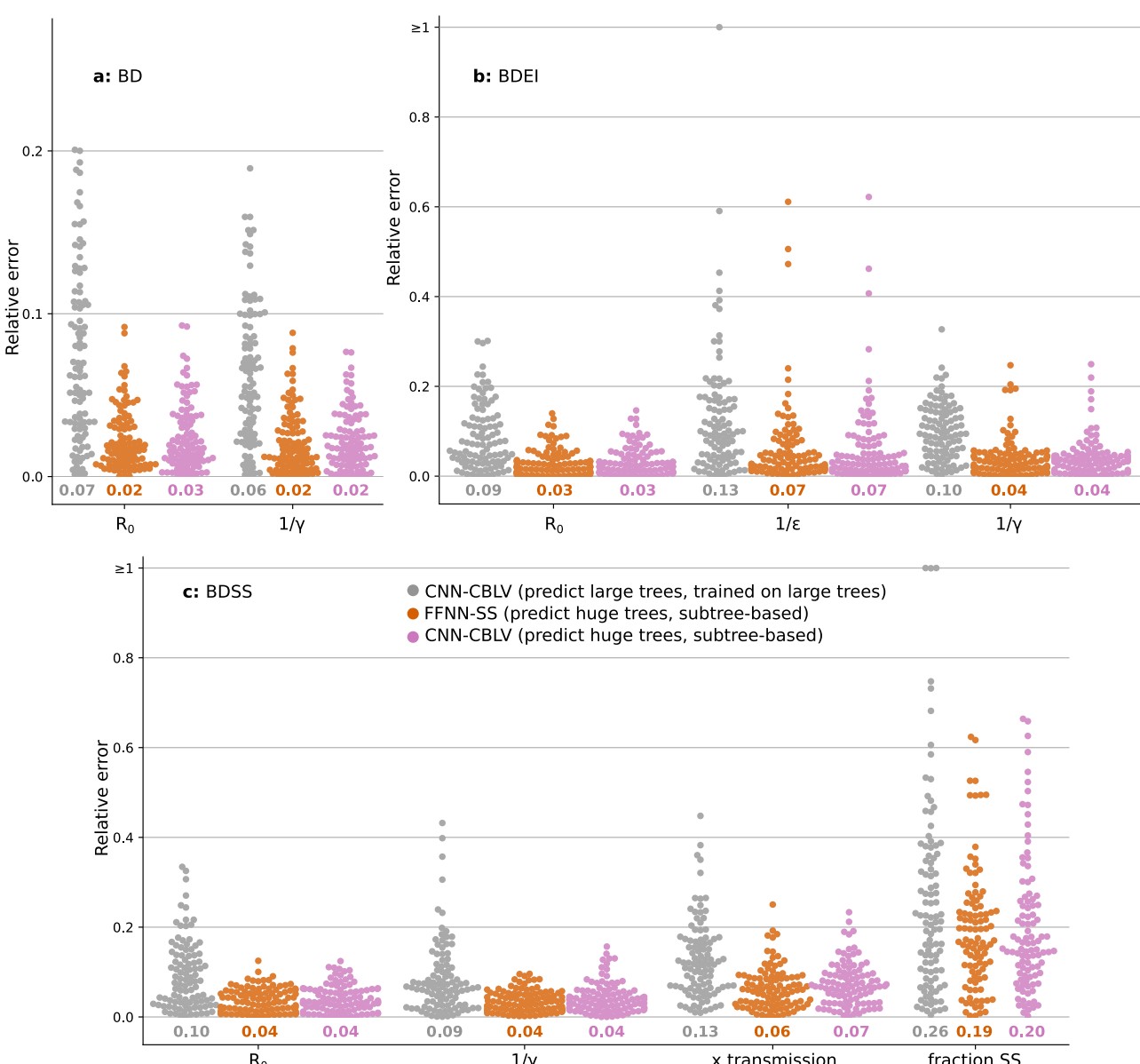

**Fig. 4 Deep learning accuracy with 'huge' trees.** Comparison of inference accuracy by neural networks trained on large trees in predicting large trees (CNN-CBLV, in grey, same as in Fig. 3) and huge trees (FFNN-SS, in orange, and CBLV-NN, in pink) on 100 large and 100 huge test trees. The training and testing large trees are the same as in Fig. 3 (between 200 and 500 tips each). The huge testing trees were generated for the same parameters as the large training and testing trees, but their size varied between 5000 and 10,000 tips. We show the relative error for each test tree. The error is measured as the normalized distance between the point estimates by neural networks and the target values for each parameter. We compare the relative errors for **a** BD-simulated, **b** BDEI-simulated and **c** BDSS-simulated trees. Average relative error is displayed for each parameter and method in corresponding colour below each plot.

not be appropriate depending on the model and the summary statistics. Moreover, rejection has a high computational cost with large simulation sets.

Neural networks circumvent these problems with rejection and allow for more complex, non-linear relationships between the tree representation and the inferred values to be captured. This is also reflected in our results with FFNN-CBLV and CNN-CRV, which both proved to be generally more accurate than LR-SS. However, FFNN-CBLV was substantially less accurate than CNN-CBLV (Supplementary Table 1, Supplementary Fig. 5). This indicates the presence of repeated patterns that may appear all along the vectorial representation of trees, such as subtrees of any size, which are better extracted by CNN than by FFNN. In its turn, CNN-CRV required larger training sets to reach an accuracy

comparable to CNN-CBLV (Supplementary Table 1, Supplementary Fig. 5), showing that the ladderization and bijectivity of the CBLV helped the training.

To assess how much information is actually learned, we also measured the accuracy of two "null models": FFNN trained to predict randomly permuted target values; and a random predictor, where parameter values were sampled from prior distributions. Results show that the neural networks extract a considerable amount of information for most of the estimated parameters (Supplementary Table 1). The most difficult parameter to estimate was the fraction of superspreading individuals in BDSS model, with accuracy close to random predictions with small trees, but better performance as the tree size increases (Fig. 4, Supplementary Fig. 3).

**SS is simpler, but CBLV has high potential for application to new models**. FFNN-SS and CNN-CBLV show similar accuracy across all settings (Fig. 3, Supplementary Tables 1, 2), including when predicting huge trees from their subtrees (Fig. 4). The only exception is the prediction of large trees using NNs trained with small trees (Supplementary Fig. 4), where FFNN-SS is superior to CNN-CBLV, but this goes beyond the recommended use of the approach, as only a part of the (large) query tree is given to the (small) CNN-CBLV.

However, the use of the two representations is clearly different, and it is likely that with new models and scenarios their accuracy will differ. SS requires a simpler architecture (FFNN) and is trained faster (e.g., 5 h with large BDSS trees), with less training instances (Supplementary Fig. 6). However, this simplicity is obtained at the cost of a long preliminary work to design appropriate summary statistics for each new model, as was confirmed in our analyses of BDSS simulations. To estimate the parameters of this model, we added summary statistics on transmission chains on top of the SS taken from Saulnier et al.[19]. This improved the accuracy of superspreading fraction estimates of the FFNN-SS, so that it was comparable to the CNN-CBLV, while the accuracy for the other parameters remained similar (Supplementary Fig. 7). The advantage of the CBLV is its generality, meaning there is no loss of information between the tree and its representation in CBLV regardless of which model the tree was generated under. However, CBLV requires more complex architectures (CNN), more computing time in the learning phase (150 h with large BDSS trees) and more training instances (Supplementary Fig. 6). Such an outcome is expected. With raw CBLV representation, the convolutional architecture is used to "discover" relevant summary statistics (or features, in machine learning terminology), which has a computational cost.

In fact, the two representations should not be opposed. An interesting direction for further research would be to combine them (e.g. during the FFNN phase), to possibly obtain even better results. Moreover, SS are still informative and useful (and quickly computed), in particular to perform sanity checks, both a priori and a posteriori (Fig. 5, Supplementary Fig. 8), or to quickly evaluate the predictability of new models and scenarios.

**Showcase study of HIV in MSM subpopulation in Zurich**. The Swiss HIV Cohort is densely sampled, including more than 16,000 infected individuals[24]. Datasets extracted from this cohort have often been studied in phylodynamics[8,25]. We analysed a dataset of an MSM subpopulation from Zurich, which corresponds to a cluster of 200 sequences studied previously by Rasmussen et al.[25], who focused on the degree of connectivity and its impact on transmission between infected individuals. Using coalescent approaches, they detected the presence of highly connected individuals at the beginning of the epidemic and estimated $R_0$ to be between 1.0 and 2.5. We used their tree as input for neural networks and BEAST2.

To perform analyses, one needs an estimate of the sampling probability. We considered that: (1) the cohort is expected to include around 45% of Swiss individuals infected with HIV[24]; and (2) the sequences were collected from around 56% of individuals enroled in this cohort[34]. We used these percentages to obtain an approximation of sampling probability of $0.45*0.56 \sim 0.25$ and used this value to analyse the MSM cluster. To check the robustness of our estimates, we also used sampling probabilities of 0.2 and 0.3 in our estimation procedures.

First, we performed a quick sanity check considering the resemblance of HIV phylogeny with simulations obtained with each model. Two approaches were used, both based on SS (Supplementary Fig. 8). Using principal component analysis (PCA), all three considered birth-death models passed the check. However, when looking at the 98 SS values in detail, namely checking whether the observed tree SS were within the [min, max] range of the corresponding simulated values, the BD and BDEI models were rejected for some of the SS (5 for both models, all related to branch lengths). Then, we performed model selection (BD vs BDEI vs BDSS) and parameter estimation using our two methods and BEAST2 (Fig. 5a, b). Finally, we checked the model adequacy with a second sanity check, derived from the inferred values and SS (Fig. 5c, Supplementary Fig. 8).

Model selection with CNN-CBLV and FFNN-SS resulted in the acceptance of BDSS (probability of 1.00 versus 0.00 for BD and BDEI), and the same result was obtained with BEAST2 and AICM. These results are consistent with our detailed sanity check, and with what is known about HIV epidemiology, namely, the presence of superspreading individuals in the infected subpopulation[35] and the absence of incubation period without infectiousness such as is emulated in BDEI[36].

We then inferred parameter values under the selected BDSS model (Fig. 5a, b). The values obtained with FFNN-SS and CNN-CBLV are close to each other, and the 95% CI are nearly identical. We inferred an $R_0$ of 1.6 and 1.7, and an infectious period of 10.2 and 9.8 years, with FFNN-SS and CNN-CBLV, respectively. Transmission by superspreading individuals was estimated to be around 9 times higher than by normal spreaders and superspreading individuals were estimated to account for around 7–8% of the population. Our $R_0$ estimates are consistent with the results of a previous study[8] performed on data from the Swiss cohort, and the results of Rasmussen et al.[25] with this dataset. The infectious period we inferred is a bit longer than that reported by Stadler et al., who estimated it to be 7.74 [95% CI 4.39–10.99] years[8]. The infectious period is a multifactorial parameter depending on treatment efficacy and adherence, the times from infection to detection and to the start of treatment, etc. In contrast to the study by Stadler et al., whose data were sampled in the period between 1998 and 2008, our dataset covers also the period between 2008 and 2014, during which life expectancy of patients with HIV was further extended[37]. This may explain why we found a longer infectious period (with compatible CIs). Lastly, our findings regarding superspreading are in accordance with those of Rassmussen et al.[25], and with a similar study in Latvia[5] based on 40 MSM sequences analysed using a likelihood approach. Although the results of the latter study may not be very accurate due to the small dataset size, they still agree with ours, giving an estimate of a superspreading transmission ratio of 9, and 5.6% of superspreading individuals. Our estimates were quite robust to the choice of sampling probability (e.g., $R_0 = 1.54$, 1.60 and 1.66, with FFNN-SS and a sampling probability of 0.20, 0.25 and 0.30, respectively, Fig. 5b).

Compared to BEAST2, the estimates of the infectious period and $R_0$ were similar for both approaches, but BEAST2 estimates were higher for the transmission ratio (14.5) and the superspreading fraction (10.6%). These values are in accordance with the positive bias of BEAST2 estimates that we observed in our simulation study for these two parameters, while our estimates were nearly unbiased (Supplementary Table 2).

Finally, we checked the adequacy of BDSS model by resemblance of HIV phylogeny to simulations. Using inferred 95% CI, we simulated 10,000 trees and performed PCA on SS, to which we projected the SS of our HIV phylogeny. This was close to simulations, specifically close to the densest swarm of simulations, supporting the adequacy of both the inferred values and the selected model (Fig. 5c). When looking at the 98 SS in detail, some of the observed values where not in the [min, max] range of the 10,000 simulated values. However, these discordant SS were all related to the lineage-through-time plot (LTT; e.g.,

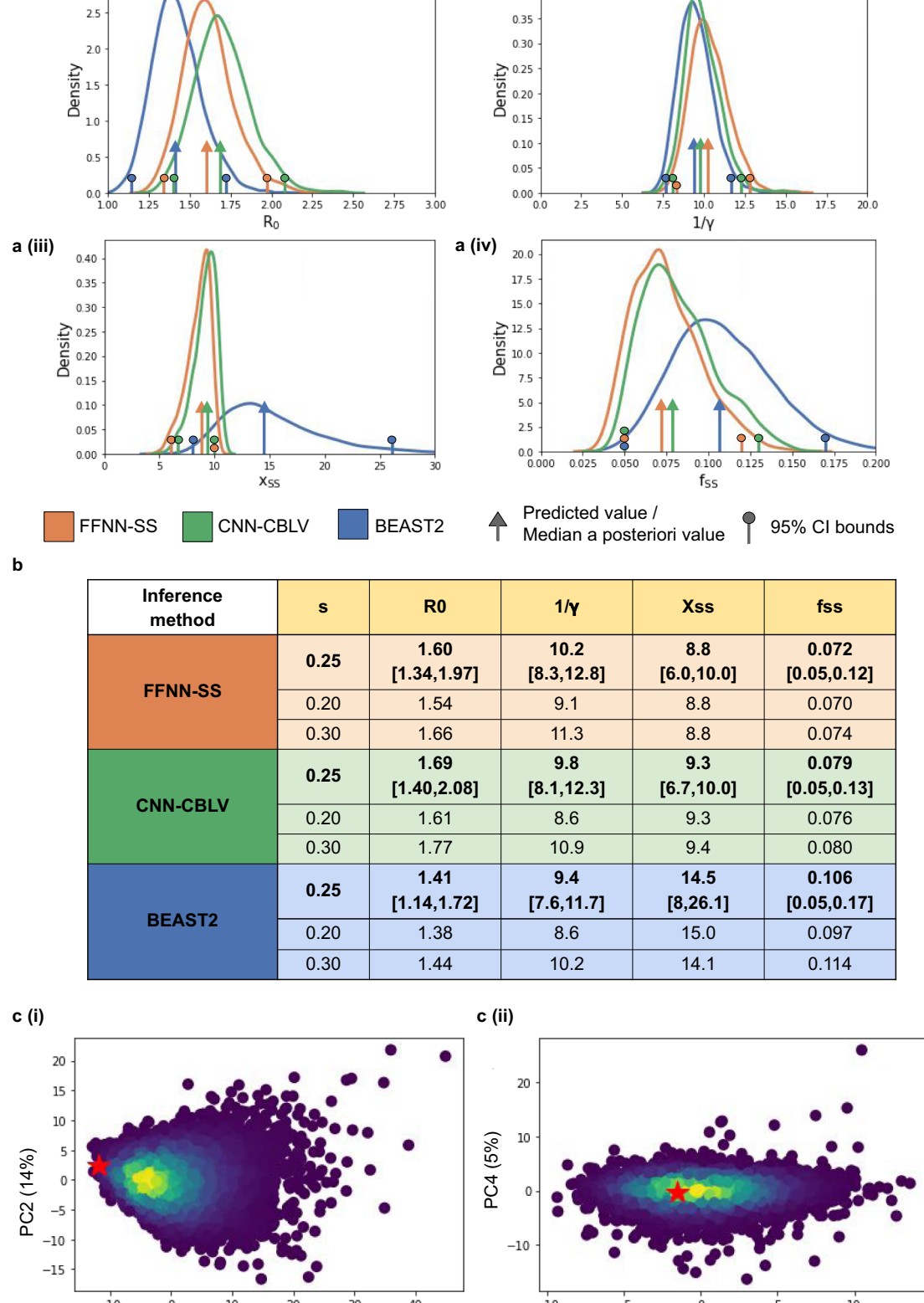

| Inference method | s | R0 | 1/γ | Xss | fss |
|---|---|---|---|---|---|
| **FFNN-SS** | 0.25 | **1.60**<br>**[1.34,1.97]** | **10.2**<br>**[8.3,12.8]** | **8.8**<br>**[6.0,10.0]** | **0.072**<br>**[0.05,0.12]** |
| | 0.20 | 1.54 | 9.1 | 8.8 | 0.070 |
| | 0.30 | 1.66 | 11.3 | 8.8 | 0.074 |
| **CNN-CBLV** | 0.25 | **1.69**<br>**[1.40,2.08]** | **9.8**<br>**[8.1,12.3]** | **9.3**<br>**[6.7,10.0]** | **0.079**<br>**[0.05,0.13]** |
| | 0.20 | 1.61 | 8.6 | 9.3 | 0.076 |
| | 0.30 | 1.77 | 10.9 | 9.4 | 0.080 |
| **BEAST2** | 0.25 | **1.41**<br>**[1.14,1.72]** | **9.4**<br>**[7.6,11.7]** | **14.5**<br>**[8,26.1]** | **0.106**<br>**[0.05,0.17]** |
| | 0.20 | 1.38 | 8.6 | 15.0 | 0.097 |
| | 0.30 | 1.44 | 10.2 | 14.1 | 0.114 |

$x$ and $y$ coordinates of this plot; Supplementary Fig. 8), consistent with the fact that the probabilistic, sampling component of the BDSS model is an oversimplification of actual sampling schemes, which depend on contact tracing, sampling campaigns and policies, etc.

## Discussion

In this manuscript, we presented new methods for parameter inference and model selection in phylodynamics based on deep learning from phylogenies. Through extensive simulations, we established that these methods are at least as accurate

**Fig. 5 Parameter inference on HIV data sampled from MSM in Zurich.** Using BDSS model with BEAST2 (in blue), FFNN-SS (in orange), and CNN-CBLV (in green) we infer: **a** (i) basic reproduction number, **a** (ii) infectious period (in years), **a** (iii) superspreading transmission ratio, and **a** (iv) superspreading fraction. For FFNN-SS and CNN-CBLV, we show the posterior distributions and the 95% CIs obtained with a fast approximation of the parametric bootstrap ('Methods', Supplementary Information). For BEAST2, the posterior distributions and 95% CI were obtained considering all reported steps (9000 in total) excluding the 10% burn-in. Arrows show the position of the original point estimates obtained with FFNN-SS and CNN-CBLV and the median a posteriori estimate obtained with BEAST2. Circles show lower and upper boundaries of 95% CI. **b** These values are reported in a table, together with point estimates obtained while considering lower and higher sampling probabilities (0.20 and 0.30). **c** 95% CI boundaries obtained with FFNN-SS are used to perform an a posteriori model adequacy check. We simulated 10,000 trees with BDSS while resampling each parameter from a uniform distribution, whose upper and lower bounds were defined by the 95% CI. We then encoded these trees into SS, performed PCA and projected SS obtained from the HIV MSM phylogeny (red stars) on these PCA plots. We show here the projection into **c** (i) first two components of PCA, **c** (ii) the 3rd and 4th components, together with the associated percentage of variance displayed in parentheses. Warm colours correspond to high density of simulations.

as state-of-the-art methods and capable of predicting very large trees in minutes, which cannot be achieved today by any other existing method. We also applied our deep learning methods to the Swiss HIV dataset from MSM and obtained results consistent with current knowledge of HIV epidemiology.

Using BEAST2, we obtained inaccurate results for some of the BDEI and BDSS simulations. While BEAST2 has been successfully deployed on many models and tasks, it clearly suffers from approximations in likelihood computation with these two models. However, these will likely improve in near future. In fact, we already witnessed substantial improvements done by BEAST2 developers to the BDSS model, while carrying out this research.

Both of our neural network approaches circumvent likelihood computation and thereby represent a new way of using molecular data in epidemiology, without the need to solve large systems of differential equations. This opens the door to novel phylodynamics models, which would make it possible to answer questions previously too complex to ask. This is especially true for CBLV representation, which does not require the design of new summary statistics, when applied to trees generated by new mathematical models. A direction for further research would be to explore such models, for example based on structured coalescent[38,39], or to extend the approach to macroevolution and species diversification models[40], which are closely related to epidemiological models. Other fields related to phylodynamics, such as population genetics, have been developing likelihood-free methods[41], for which our approach might serve as a source of inspiration.

A key issue in both phylodynamics and machine learning applications is scalability. Our results show that very large phylogenies can be analysed very efficiently (~1 min for 10,000 tips), with resulting estimates more accurate than with smaller trees (Fig. 4), as predicted by learning theory. Again, as expected, more complex models require more training instances, especially BDSS using CBLV (Supplementary Fig. 3), but the ratio remains reasonable, and it is likely that complex (but identifiable) models will be handled efficiently with manageable training sets. Surprisingly, we did not observe a substantial drop of accuracy with lower sampling probabilities. To analyse very large trees, we used a decomposition into smaller, disjoint subtrees. In fact, all our NNs were trained with trees of moderate size (<500 tips). Another approach would be to learn directly from large trees. This is an interesting direction for further research, but this poses several difficulties. The first is that we need to simulate these very large trees, and a large number of them (millions or more). Then, SS is the easiest representation to learn, but at the risk of losing essential information, which means that new summary statistics will likely be needed for sufficiently complete representation of very large phylogenies. Similarly, with CBLV more complex NN architectures (e.g., with additional and larger kernels in the convolutional layers) will likely be needed, imposing larger training sets. Combining both representations (e.g., during the

FFNN phase) is certainly an interesting direction for further research. Note, however, that the predictions of both approaches for the three models we studied are highly correlated (Pearson coefficient nearly equal to 1 for most parameters), which means that there is likely little room for improvement (at least with these models).

A key advantage of the deep learning approaches is that they yield close to immediate estimates and apply to trees of varying size. Collection of pathogen genetic data became standard in many countries, resulting in densely sampled infected populations. Examples of such datasets include HIV in Switzerland and UK[24,42], 2013 Ebola epidemics[6], several Influenza epidemics and the 2019 SARS-Cov-2 pandemic (www.gisaid.org)[43]. For many such pathogens, trees can be efficiently and accurately inferred[44–46] and dated[47–49] using standard approaches. When applied to such dated trees, our methods can perform model selection and provide accurate phylodynamic parameter estimates within a fraction of a second. Such properties are desirable for phylogeny-based real-time outbreak surveillance methods, which must be able to cope with the daily influx of new samples, and thus increasing size of phylogenies, as the epidemic unfolds, in order to study local outbreaks and clusters, and assess and compare the efficiency of healthcare policies deployed in parallel. Moreover, thanks to the subtree picking and averaging strategy, it is now possible to analyse very large phylogenies, and the approach could be used to track the evolution of parameters (e.g., $R_0$) in different regions (sub-trees) of a global tree, as a function of dates (as in Bayesian skyline models[4]), geographical areas, viral variants etc.

## Methods

Here we describe the main methodological steps. For algorithms, technical details, software programs used and their options, and additional comments, an extended version is available in Supplementary Information.

**Tree representation using summary statistics (SS)**. We use 98 summary statistics (SS), to which we add the sampling probability, summing to a vector of 99 values. We use the 83 SS proposed by Saulnier et al.[19]:

- 8 SS on tree topology.
- 26 SS on branch lengths.
- 9 SS on lineage-through-time (LTT) plot.
- 40 SS providing the coordinates of the LTT plot.

In addition, we designed 14 SS on transmission chains to capture information on the superspreading population. A superspreading individual transmits to more individuals within a given time period than a normal spreader. We thus expect that with superspreading individuals we would have shorter transmission chains. To have a proxy for the transmission chain length, we look at the sum of 4 subsequent shortest times of transmission for each internal node. This gives us a distribution of time-durations of 4-transmission chains. We assume that information on the transmission dynamics of superspreading individuals is retained in the lower (i.e., left) tail of this distribution, which contains relatively many transmissions with short time to next transmission, while the information on normal spreaders should be present in the rest of the distribution. From the observed distribution of 4-transmission-chain lengths, we compute 14 statistics:

- Number of 4-transmission chains in the tree.
- 9 deciles of 4-transmission-chain lengths distribution.
- Minimum and maximum values of 4-transmission-chain lengths.
- Mean value of 4-transmission-chain lengths.
- Variance of 4-transmission-chain lengths.

Moreover, we provide the number of tips in the input tree, resulting in $83 + 14 + 1 = 98$ SS in total.

**Complete and compact tree representation (CBLV)**. The representation of a tree with $n$ tips is a vector of length $2n-1$, where one single real-valued scalar corresponds to one internal node or tip. This representation thus scales linearly with the tree size. The encoding is achieved in two steps: tree ladderization and tree traversal.

The tree ladderization consists in ordering the children of each node. Child nodes are sorted based on the sampling time of the most recently sampled tip in their subtrees: for each node, the branch supporting the most recently sampled subtree is rotated to the left, as in Fig. 2a (i–ii).

Once the tree is sorted, we perform an inorder tree traversal[30]. When visiting a tip, we add its distance to the previously visited internal node or its distance to the root, for the tip that is visited first (i.e., the tree height due to ladderization). When visiting an internal node, we add its distance to the root. Examples of encoding are shown in Fig. 2a (ii–iii). This gives us the Compact Bijective Ladderized Vector (CBLV). We then separate information relative to tips and to internal nodes into two rows (Fig. 2a (iv)) and complete the representation with zeros until reaching the size of the largest simulated tree for the given simulation set (Fig. 2a (v)).

CBLV has favourable features for deep learning. Ladderization does not actually change the input tree, but by ordering the subtrees it standardizes the input data and facilitates the learning phase, as observed with random subtree order (Supplementary Fig. 5, Compact Random Vector (CRV) representation,). The inorder tree traversal procedure is a bijective transformation, as it transforms a tree into a tree-compatible vector, from which the (ordered) tree can be reconstructed unambiguously, using a simple path-agglomeration algorithm shown in Supplementary Fig. 1. CBLV is "as concise as possible". A rooted tree has $2n-2$ branches, and thus $2n-2$ entries are needed to represent the branch lengths. In our $2n-1$ vectorial encoding of trees, we not only represent the branch lengths, but also the tree topology using only 1 additional entry.

**Tree rescaling**. Before encoding, the trees are rescaled so that the average branch length is 1, that is, each branch length is divided by the average branch length of the given tree, called rescale factor. The values of the corresponding time-dependent parameters (i.e., infectious period and incubation period) are divided by the rescale factor too. The NN is then trained to predict these rescaled values. After parameter prediction, the predicted parameter values are multiplied by the rescale factor and thus rescaled back to the original time scale. Rescaling thus makes a pre-trained NN more generally applicable, for example both to phylogenies of pathogen-associated with an infectious period on the scale of days (e.g., Ebola virus) and years (e.g., HIV).

**Reduction and centering of summary statistics**. Before training our NN and after having rescaled the trees to unit average branch length (see above), we reduce and centre every summary statistic by subtracting the mean and scaling to unit variance. To achieve this, we use the standard scaler from the scikit-learn package[50], which is fitted to the training set. This does not apply to CBLV representation, to avoid losing the ability to reconstruct the tree.

**Parameter and model inference using neural networks**. We implemented deep learning methods in Python 3.6 using Tensorflow 1.5.0[51], Keras 2.2.4[52] and scikit-learn 0.19.1[50] libraries. For each network, several variants in terms of number of layers and neurons, activation functions, regularization, loss functions and optimizer, were tested. In the end, we decided for two specific architectures that best fit our purpose: one deep FFNN trained on SS and one CNN trained on CBLV tree representation.

The FFNN for SS consists of one input layer with 99 input nodes (98 SS + the sampling probability), 4 sequential hidden layers organized in a funnel shape with 64-32-16-8 neurons and 1 output layer of size 2–4 depending on the number of parameters to be estimated. The neurons of the last hidden layer have linear activation, while others have exponential linear activation[53].

The CNN for CBLV consists of one input layer (of 400 and 1002 input nodes for trees with 50–199 and 200–500 tips, respectively). This input is then reshaped into a matrix of size of $201 \times 2$ and $501 \times 2$, for small and large trees, respectively, with entries corresponding to tips and internal nodes separated into two different rows (and one extra column with one entry in each row corresponding to the sampling probability). Then, there are two 1D convolutional layers of 50 kernels each, of size 3 and 10, respectively, followed by max pooling of size 10 and another 1D convolutional layer of 80 kernels of size 10. After the last convolutional layer, there is a GlobalPoolingAverage1D layer and a FFNN of funnel shape (64-32-16-8 neurons) with the same architecture and setting as the NN used with SS.

For both NNs, we use the Adam algorithm[54] as optimizer and the Mean Absolute Percentage Error (MAPE) as loss function. The batch size is set to 8000.

To train the network, we split the simulated dataset into 2 groups: (1) proper training set (3,990,000 examples); (2) validation set (10,000). To prevent overfitting during training, we use: (1) the early stopping algorithm evaluating MAPE on the validation set; and (2) dropout that we set to 0.5 in the feedforward part of both NNs[55] (0.4, 0.45, 0.55 and 0.6 values were tried for basic BD model without improving the accuracy).

For model selection, we use the same architecture for FFNN-SS and CNN-CBLV as those for parameter inference described above. The only differences are: (1) the cost function: categorical cross entropy, and (2) the activation function used for the output layer, that is, softmax function (of size 2 for small trees, selecting between BD and BDEI model, and of size 3 for large trees, selecting between BD, BDEI and BDSS). As we use the softmax function, the outputs of prediction are the estimated probabilities of each model, summing to 1.

**Parameter estimation from very large trees using subtree picking and averaging**. To estimate parameters from very large trees we designed the 'Subtree Picker' algorithm (see Supplementary Information for details). The goal of Subtree Picker is to extract subtrees of bounded size representing independent sub-epidemics within the global epidemic represented by the initial huge tree T, while covering most of the initial tree branches and tips in T. The sub-epidemics should follow the same sampling scheme as the global epidemic. This means that we can stop the sampling earlier than the most recent tip in T, but we cannot omit tips sampled before the end the sampling period (this would correspond to lower sampling probability). Each picked subtree corresponds to a sub-epidemic that starts with its root individual and lasts between its root date $D_{root}$ and some later date ($D_{last} > D_{root}$). The picked subtree corresponds to the top part of the initial tree's clade with the same root, while the tips sampled after $D_{last}$ are pruned. The picked subtrees do not intersect with each other and cover most of the initial tree's branches: 98.5% (BD), 97.3% (BDEI) and 82.4% (BDSS) of the initial tree branches on the 'huge' tree datasets (5000 to 10,000 tips). For the BDSS model, this percentage is lower than for BD and BDEI, because of the narrower subtree size interval (200-to-500 tips versus 50-to-500 tips) corresponding to current Phylo-Deep training set settings. Subtree Picker performs a postorder tree traversal (tips-to-root) and requires $O(n^2)$ computing time in the worst case, where $n$ is the number of tips in T. In practice, Subtree Picker takes on average 0.6 (BD), 0.8 (BDEI) and 0.8 (BDSS) seconds per 'huge' tree (5000-to-10,000 tips), meaning that it could easily be applied to much larger trees. Once subtrees (sub-epidemics) have been extracted, they are analysed using CNN-CBLV or FFNN-SS, and the parameter estimates are averaged with weights proportional to subtree sizes.

**Confidence intervals**. For all NN-based parameter estimates, we compute 95% CI using a form of parametric bootstrap. To facilitate the deployment and speed-up the computation, we perform an approximation using a separate set of 1,000,000 simulations. For each simulation in the CI set, we store the true parameter values and the parameter values predicted with both of our methods. This large dataset of true/predicted values is used to avoid new simulations, as required with the standard parametric bootstrap. For a given simulated or empirical tree T, we obtain a set of predicted parameter values, {p}. The CI computation procedure searches among stored data those that are closest to T in terms of tree size, sampling probability and predicted values. We first subset:

- 10% of simulations within the CI set, which is closest to T in terms of size (number of tips).
- Amongst these, 10% of simulations that are closest to T in terms of sampling probability.

We thus obtain 10,000 CI sets of real/predicted parameter values, similar in size and sampling probability to T. For each parameter value $p$ predicted from T, we identify the 1000 nearest neighbouring values amongst the 10,000 true values of the same parameter available in the CI sets, $R_{CI} = \{r_{i=1,1000}\}$, and keep the corresponding predicted values, $P_{CI} = \{p_{i=1,1000}\}$. We then measure the errors for these neighbours as $E_{CI} = \{e_i = p_i - r_i\}$. We centre these errors around $p$ using the median of errors, $m(E_{CI})$, which yields the distribution of errors for given prediction $p$: $D = \{p + e_i - m(E_{CI})\}$, from which we extract the 95% CI around $p$. Individual points in the obtained distribution that are outside of the parameter ranges covered through simulations are set to the closest boundary value of the parameter range. With very large trees and the subtree picking and averaging procedure, we use a quadratic weighted average of the individual CIs found for every subtree. To assess this fast implementation of the parametric bootstrap, we used the coverage of the true parameter values (expected to be of 95%) and the width (the lower the better) of the CIs. Results and comparisons with BEAST2 are reported in Supplementary Table 7.

**Models**. The models we used for tree simulations are represented in the form of flow diagrams in Fig. 1. We simulated dated binary trees for (1) the training of NNs and (2) accuracy assessment of parameter estimation and model selection. We used the following three individual-based phylodynamic models:

- Constant rate birth-death model with incomplete sampling: This model (BD[8,9], Fig. 1a) contains three parameters and three compartments:

infectious (I), removed with sampling (R) and removed unsampled (U) individuals. Infection takes place at rate β. Infectious individuals are removed with rate γ. Upon removal, an individual is sampled with probability s. For simulations, we re-parameterized the model in terms of: basic reproduction number, $R_0$; infectious period, $1/\gamma$; sampling probability, s; and tree size, t. We then sampled the values for each simulation uniformly at random in the ranges given in Supplementary Table 4.

- Birth-death model with exposed-infectious classes: This model (BDEI[10–12], Fig. 1b) is a BD model extended through the presence of an exposed class. More specifically, this means that each infected individual starts as non-infectious (E) and becomes infectious (I) at incubation rate ε. BDEI model thus has four parameters (β, γ, ε and s) and four compartments (E, I, R and U). For simulations, we re-parameterized the model similarly as described for BD and set the ε value via the incubation ratio ($=\varepsilon/\gamma$). We sampled all parameters, including $\varepsilon/\gamma$, from a uniform distribution, just as with BD (Supplementary Table 4).

- Birth-death model with superspreading: This model (BDSS[5,10,11], Fig. 1c) accounts for heterogeneous infectious classes. Infected individuals belong to one of two infectious classes ($I_S$ for superspreading and $I_N$ for normal spreading) and can transmit the disease by giving birth to individuals of either class, with rates $\beta_{S,S}$ and $\beta_{S,N}$ for $I_S$ transmitting to $I_S$ and to $I_N$, respectively, and $\beta_{N,S}$ and $\beta_{N,N}$ for $I_N$ transmitting to $I_S$ and $I_N$, respectively. However, there is a restriction on parameter values: $\beta_{S,S} \times \beta_{N,N} = \beta_{S,N} \times \beta_{N,S}$. There are thus superspreading transmission rates $\beta_S$, and normal transmission rates $\beta_N$, that are $X_{SS} = \beta_{S,S}/\beta_{N,S} = \beta_{S,N}/\beta_{N,N}$ times higher for superspreading. At transmission, the probability of the recipient to be superspreading is $f_{SS} = \beta_{S,S}/(\beta_{S,S} + \beta_{S,N})$, the fraction of superspreading individuals at equilibrium. We consider that both $I_S$ and $I_N$ populations are otherwise indistinguishable, that is, both populations share the same infectious period $(1/\gamma)$[5,10,11]. The model thus has six parameters, but only five need to be estimated to fully define the model[5,10]. For simulations, we chose parameters of epidemiological interest for re-parameterization: basic reproduction number $R_0$, infectious period $1/\gamma$, $f_{SS}$, $X_{SS}$ and sampling probability s. In our simulations, we used uniform distributions for these five parameters, just as with BD and BDEI (Supplementary Table 4).

**Parameter inference with BEAST2.** To assess the accuracy of our methods, we compared it with a well-established Bayesian method, as implemented in BEAST2 (version 2.6.2). We used the BDSKY package[4] (version 1.4.5) to estimate the parameter values of BD simulations and the package bdmm[12,13] (version 1.0) to infer the parameter values of BDEI and BDSS simulations. Furthermore, for the inference on BDSS simulations, instead of BEAST 2.6.2 we used the BEAST2 code up to the commit nr2311ba7, which includes important fixes to operators critical for our analyses. We set the Markov Chain Monte Carlo (MCMC) length to 5 million steps for the BD model, and to 10 million steps for the BDEI and BDSS models.

The sampling probability was fixed during the estimation. Since the BD, BDEI and BDSS models implemented in BEAST2 do not use the same parametrizations as our methods, we needed to apply parameter conversions for setting the priors for BEAST2 inference (Supplementary Table 6), and for translating the BEAST2 results back to parameterizations used in our methods, in order to enable proper comparison of the results (see Supplementary Information for details).

After we obtained the parameters of interest from the original parameters estimated by BEAST2, we evaluated the Effective Sample Size (ESS) on all parameters. We reported the absolute percentage error of the median of a posteriori values (more stable and accurate than the maximum a posteriori), corresponding to all reported steps (spaced by 1000 actual MCMC steps) past the 10% burn-in. For simulations for which BEAST2 did not converge, we considered the median of the parameter distribution used for simulations (Fig. 3, Supplementary Tables 1, 2, Supplementary Fig. 2) or excluded them from the comparison (Supplementary Tables 1, 2, values reported in brackets, Supplementary Table 5).

For the HIV application, the prior of infectious period was set to [0.1, 30] years (uniform). All the other parameters had the same prior distributions as used in simulations and shown in Supplementary Tables 4, 6.

**Accuracy of parameter estimation.** To compare the accuracy of the different methods, we used 100 simulated trees per model. For each simulated tree, we computed the relative error and its mean over the 100 trees (Figs. 3–4, Supplementary Table 1, Supplementary Figs. 2–4):

$$\text{MRE} = \frac{1}{100}\sum_{i=1}^{100}\frac{|\text{predicted}_i - \text{target}_i|}{\text{target}_i}.$$

The mean relative bias (Supplementary Table 2) was measured in a similar manner as:

$$\text{MRB} = \frac{1}{100}\sum_{i=1}^{100}\frac{(\text{predicted}_i - \text{target}_i)}{\text{target}_i}.$$

**Comparison of time efficiency.** For FFNN-SS and CNN-CBLV, we reported the average CPU time of encoding a tree (average over 10,000 trees), as reported by NextFlow[56]. The inference time itself was negligible.

For BEAST2, we reported the CPU time averaged over 100 analyses with BEAST2 as reported by NextFlow. For the analyses with BDEI and BDSS models, we reported the CPU time to process 10 million MCMC steps, and for the analyses with BD, we reported the CPU time to process 5 million MCMC steps. To account for convergence, we re-calculated the average CPU time considering only those analyses for which the chain converged and an ESS of 200 was reached for all inferred parameters.

The calculations were performed on a computational cluster with CentOS machines and Slurm workload manager. The machines had the following characteristics: 28 cores, 2.4 GHz, 128 GB of RAM. Each of our jobs (simulation of one tree, tree encoding, BEAST2 run, etc.) was performed requesting one CPU core. The neural network training was performed on a GPU cluster with Nvidia Titan X GPUs.

**HIV dataset.** We used the original phylogenetic tree reconstructed by Rasmussen et al.[25] from 200 sequences corresponding to the largest cluster of HIV-infected men-having-sex-with-men (MSM) subpopulation in Zurich, collected as a part of the Swiss Cohort Study[24]. For details on tree reconstruction, please refer to their article.

**PhyloDeep software.** FFNN-SS and CNN-CBLV parameter inference, model selection, 95% CI computation and a priori checks are implemented in the PhyloDeep software, which is available on GitHub (github.com/evolbioinfo/phylodeep), PyPi (pypi.org/project/phylodeep) and Docker Hub (hub.docker.com/r/evolbioinfo/phylo-deep). It can be run as a command-line programme, Python3 package and a Docker container. PhyloDeep covers the parameter subspace as described in Supplementary Table 4. The input is a dated phylogenetic tree with at least 50 tips and presumed sampling probability. The output is a PCA plot for a priori check, a csv file with all SS, a csv file with probabilities of each model (for model selection) and point estimates and 95% CI values (for parameter inference with selected model). The installation details and usage examples are available as well on GitHub.

**Reporting summary.** Further information on research design is available in the Nature Research Reporting Summary linked to this article.

## Data availability
The data generated in this study (simulated trees, BEAST2 logs, and the results of BEAST2 and PhyloDeep runs), as well as the HIV phylogenetic tree for Zurich epidemic (a showcase application) are provided on GitHub (github.com/evolbioinfo/phylodeep, version 0.3) and have been deposited in the Zenodo database under accession code https://doi.org/10.5281/zenodo.6646668. The simulated trees were obtained with our simulator (see Code availability), the HIV tree was previously published by Rasmussen et al.[25] and is available on their GitHub: github.com/davidrasm/PairTree (all confidential information has been removed).

## Code availability
The PhyloDeep package (version 0.3) is under the GPL v3.0 license and uses Python (3.6) and Python libraries: ete3 (version 3.1.2 under GNU general licence); pandas (version 1.1.5); numpy (version 1.19.5); scipy (version 1.1.0); scikit-learn (version 0.19.1); tensorflow (version 1.13.1); joblib (version 0.13.2); h5py (version 2.10.0); Keras (version 2.4.3 under Apache 2.0 license); matplotlib (version 3.1.3 under PSF license). We provide (i) the source code of PhyloDeep, (ii) the code of the tree simulators used to train the deep learners and (iii) the log files obtained with BEAST2 on GitHub (github.com/evolbioinfo/phylodeep). The code has been deposited in Zenodo[57]. We used the version 2.6.2 of BEAST2 for BD and BDEI inferences and BEAST2 compiled up to the commit nr2311ba7 for BDSS inferences, and version 3.3 of TreePar. For BEAST2 inferences, we used BEAST2 libraries bdmm (version 1.0) and BDSKY (version 1.4.5). We used Snakemake (version 5.10.0) and Nextflow (version 21.04.3.5560) pipeline managers for simulations and analyses of simulated data.

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

## Acknowledgements

We would like to thank Dr. Kary Ocaña and Tristan Dot for initiating experiments on machine learning and phylogenetic trees in our laboratory. We would like to thank Quang Tru Huynh for administrating the GPU farm at Institut Pasteur and the INCEPTION programme (Investissement d'Avenir grant ANR-16-CONV-0005) that financed the GPU farm. We would like to thank Dr. Christophe Zimmer from Institut Pasteur, Sophia Lambert and Dr. Hélène Morlon from Institut de Biologie de l'Ecole Normale Supérieure IBENS and Dr. Guy Baele from Katholieke Universiteit KU Leuven for useful discussions, and Dr. Isaac Overcast from IBENS and Luc Blassel from Institut Pasteur for critical reading of the manuscript. We would like to thank Dr. Tanja Stadler and Jérémie Sciré for their help with BEAST2 and MLE approaches. J.V. is supported by Ecole Normale Supérieure Paris-Saclay and by ED Frontières de l'Innovation en Recherche et Education, Programme Bettencourt. V.B. would like to thank Swiss National Science Foundation for funding (Early PostDoc mobility grant P2EZP3_184543). O.G. is supported by PRAIRIE (ANR-19-P3IA-0001).

## Author contributions

J.V., A.Z. and O.G. conceived and set up the methods; J.V., A.Z. and V.B. performed the experiments; J.V., A.Z. and O.G. analysed the results; J.V. and A.Z. wrote the Python package; J.V. and O.G. wrote the manuscript; J.V., A.Z., V.B. and O.G. edited the

manuscript; all authors helped in this research, discussed the results and read the final manuscript; O.G. initiated and supervised the project.

## Competing interests

The authors declare no competing interests.
