## [Peer Review File · Nature Communications]

Reviewers' Comments:

Reviewer #1:

Remarks to the Author:

This paper introduces PhyloDeep, a deep learning tool for predicting phylodynamic parameters under three birth-death models. The paper introduces machine learning models based on two different representations of timed phylogenies: (1) a summary statistic based representation, which represents timed phylogenies based on extended feature set first described by Saulnier et al., and (2) a vector based representation that uniquely describes timed phylogenies, which are rooted, binary, edge-labeled trees. On simulated data of modest size (of 50-200 and 200-500 leaves), the paper demonstrates that (1) the two representations have equivalent performance for all birth-death models, (2) have similar performance as BEAST2 for the simplest model, (3) but better performance than BEAST2 for the more complicated birth-death models. Finally, the paper illustrates a use case of PhyloDeep on a HIV datasets with 200 leaves, showing that the predicted birth-death model parameters are more in line with another previous study than BEAST's. Overall, the paper is well written and thorough, and will be of interest to the readership of this journal. I have a couple of comments that could strengthen the manuscript.

1. How do learned models generalize?

I would like to know how well learned models generalize to other smaller datasets. For instance, does the learned model for the larger instances (200-500 tips) perform well on the smaller instances (50-200 tips)?

2. Guidance on which representation to use?

Related to above, I'd like to see a more thorough investigation of the differences between the two representations? Can you find scenarios where one representation would be preferable over the other? Does one take longer to train? Is there a difference in generalizability?

3. What is the impact of the additional features added to the SS representation.

The authors introduced new summary statistics. I'd like to know more about the performance with and without these additional features. Also, how well does a joint representation perform, where you combine the two current representations (based on summary statistics, and based on tree topology/branch lengths).

4. Compute likelihood of solutions identified by PhyloDeep

To better understand where the improvement of performance relative to BEAST2 comes from, it would be good to evaluate the likelihood of the solutions identified by PhyloDeep. Do they have larger likelihood than BEAST2 solutions?

5. Additional real dataset. Preferably a large-scale one.

Finally, I think the impact of the method/paper can be largely increased if you would consider an additional large-scale dataset.

Minor comments:

* More emphasis on PhyloDeep

The first mention of PhyloDeep occurs very late in the paper, almost like an afterthought. I think it should be featured more prominently, e.g. in the Abstract and Introduction, especially if you want this to be a tool to be used by the community.

* Line 452: maximal => maximum or largest

Reviewer #2:
Remarks to the Author:
Review

Deep learning from phylogenies to uncover the transmission dynamics of epidemics

By Voznica et al.

The authors propose an original likelihood-free, simulation-based approach grounded on deep learning to infer the dynamics of epidemics from genetic data. They compare different versions of their approach to existing approaches currently used in the domain. They apply the method to an existing real data set dealing with the HIV epidemic in Zurich, which was already analyzed and leads the authors to refine the knowledge about the determinants of HIV transmission in Zurich. The short discussion gives several perspectives for extending the application domain of their approach. Codes implementing the approach are provided within the Python package PhyloDeep (note that I have not tested the codes).

This study is particularly well designed and many aspects are explored (most of the interrogations that I had during the reading of the main text are actually treated in the supplementary material). The main text is clear and, as I write above, generally adequately complemented by supporting information. The methods are well described and relevant.

The following concerns more specific points.

The title, DEEP LEARNING FROM PHYLOGENIES TO UNCOVER THE TRANSMISSION DYNAMICS OF EPIDEMICS, induces a confusion since the inference of the transmission dynamics may refer to the estimation of 'who infected whom', whereas the authors' objective is upstream: selecting a transmission model and estimating its parameters. Hence, I wonder if a title like DEEP LEARNING FROM PHYLOGENIES TO INFER THE DETERMINANTS OF DISEASE TRANSMISSION DYNAMICS (or something approaching) would be more adequate.

To facilitate the reading across this rich piece of work, supplementary figures and tables should be ordered as they appear in the main text.

After reading the introduction, I was unsure whether you were embedding your method in the framework of ABC or not. If I understood correctly, you do not, and you simply mention ABC, and more specifically the paper by Saulnier et al., because you recycle summary statistics that were proposed by Saulnier and her colleagues. At l.68, the term 'rejection-free' puts the reader on the track that you do not develop an ABC approach, but you should make it more explicit at the transition from the paragraph about the Saulnier's paper (l.60-67) to the next paragraph (l.68-80).

l.122: The authors should more exactly specify here what they call the 'sampling probability' and what means 'known'. This is clear later in the paper, but is ambiguous at this stage of the paper.

l.196-197: The convergence issue only concerns BEAST2, doesn't it?

In the discussion, the authors should evoke the question about how their approaches scale up with larger data sets than those considered in the application and simulations, with larger trees, lower sampling probability, and models with more parameters. In particular, is there a need for a much larger number of simulations (than 4M) for training the deep learning tools that the authors proposed to use in these cases? Supp. Fig. 3 and 4 partly tackle this question and, if my interpretation is correct, SF3 states that the performance is relatively stable in terms of model selection when the tree size increases, and SF4 states that the performance in terms of parameter estimation accuracy increases with the number of simulations. The authors could make a synthesis of this type of results in the discussion and extrapolate (to some extents) for answering the other dimensions of the above-mentioned question.

Fig. 4 and Supp. Fig. 8: the authors interestingly show that observed summary statistics for HIV are within the 'simulated envelope' of summary statistics throughout an analysis of the first two axes of a PCA. It would be interesting as well to perform the a priori check for the row summary statistics (without PCA). Since there are many summary statistics, the authors could provide a relatively concise table indicating, for each SS (i.e., marginally), to which quantile the observed value corresponds. This table could also be summarized into an histogram providing the distribution of the afore-mentioned quantiles. The table and the histogram would more precisely indicate how the class of used models represent real data.

Supp. Table 1 should include the information given at l.823-825 that a different prior is used for the infectious period in the numerical experiment and in the application. Looking at Fig. 4, it seems to me that a different prior is also used for $X_{\{SS\}}$, but I am maybe wrong and I maybe missed this information in the text.

Samuel Soubeyrand, INRAE, BioSP.

April 22, 2022

Dear R ,

We would like to thank you for your comments on our manuscript "Deep learning from phylogenies to uncover the transmission dynamics of epidemics", submitted to Nature Communications. We have uploaded a revised version to the journal's website. We apologize for the delay in reviving this manuscript. It was a complicated time for all of us with Covid-19 and Jakub Voznica (first author) moving to another institution after his thesis defence.

Your comments helped us to improve the methods, the PhyloDeep software and the original manuscript. Following the comments of referee 1, we have considerably extended the range of application of our neural networks, making them capable of analysing very large phylogenies in a few minutes, thanks to a novel decomposition of large pathogen phylogenies into sub-epidemics (sub-trees). We also assessed the generalization capabilities and the likelihood performance of our approach. Following the comments of referee 2, the discussion has been completed and we changed the title to "Deep learning from phylogenies to infer the epidemiological dynamics of outbreaks", which is clearer.

We are confident that the new version is much improved thanks to your comments, all of which have been taken into account. In what follows, you will find our point-by-point responses/changes, as well as a highlighted version of the main manuscript where all changes are marked in blue. We have also uploaded highlighted versions of the Methods and Supplementary Information onto the journal website.

We look forward to any further comments you may have.

Sincerely, the authors

REVIEWER COMMENTS

Reviewer #1 (Remarks to the Author):

This paper introduces PhyloDeep, a deep learning tool for predicting phylodynamic parameters under three birth-death models. The paper introduces machine learning models based on two different representations of timed phylogenies: (1) a summary statistic based representation, which represents timed phylogenies based on extended feature set first described by Saulnier et al., and (2) a vector based representation that uniquely describes timed phylogenies, which are rooted, binary, edge-labeled trees. On simulated data of modest size (of 50-200 and 200-500 leaves), the paper demonstrates that (1) the two representations have equivalent performance for all birth-death models, (2) have similar performance as BEAST2 for the simplest model, (3) but better performance than BEAST2 for the more complicated birth-death models. Finally, the paper illustrates a use case of PhyloDeep on a HIV datasets with 200 leaves, showing that the predicted birth-death model parameters are more in line with another previous study than BEAST's. Overall, the paper is well written and thorough, and will be of interest to the readership of this journal. I have a couple of comments that could strengthen the manuscript.

1. How do learned models generalize?

I would like to know how well learned models generalize to other smaller datasets. For instance, does the learned model for the larger instances (200-500 tips) perform well on the smaller instances (50-200 tips)?

In statistical learning theory [31], generalization relates to the ability to predict new samples drawn from the same distribution as the training instances. Generalization is opposed to rote learning and overfitting, where the learned classifier or regressor predicts the training instances accurately, but new instances extracted from the same distribution or population poorly. The generalization capability of our NNs was extensively assessed in the submitted version of the manuscript, using large, independent testing sets (Fig. 3).

However, we agree with the referee that extending the study to samples that differ from the training distribution is clearly of interest in phylodynamics, in particular when the input tree is smaller than the training trees (as he/she suggested), but also, most importantly, when the input tree is larger than the training trees. We added results along this line. To summarize:

- We estimated the parameters of small trees (50-199 tips) using NNs trained with large trees (200-500 tips), and vice versa the parameters of large trees with NNs trained with small trees. The results (Supplementary Fig. 4) were surprisingly good from a machine learning standpoint, as the testing trees clearly departed from the training distribution. In particular, the accuracy obtained with FFNN-SS (summary statistics) was not affected very much by this strong violation of standard machine learning assumptions, while the accuracy of CNN-CBLV (combinatorial tree representation) was impacted but remained relatively high.

- However, in these experiments all trees are still of moderate size (≤ 500 tips), while very large trees will become increasingly common in the near future with viral pathogens (see the current epidemics...). We thus explored another use of our NNs (pages 7, 9-10, Fig. 4), where a 'huge' input tree (5,000 to 10,000 tips in our experiments) is first decomposed into a set of disjoint subtrees (50 to 500 tips), which cover most of the huge-tree branches. Then, we apply the NNs for predictions on each subtree and combine the results using weighted averages. The results are impressive, for both FFNN-SS and CNN-CBLV. The prediction requires ~ 1 CPU minute and the accuracy obtained with these huge trees is clearly higher than the one obtained with large trees (200-500 tips), with an error drop of a factor of 2 to 3 (Fig. 4). When applying this decomposition method to the prediction of large trees using NNs trained with small trees, the error became nearly identical to the error obtained with the right NNs (Supplementary Fig. 4).
- We believe that this capacity of NNs, made possible by their predictive speed, opens the way to many applications, which cannot be addressed today by any existing method. In particular, it is now possible to analyse extremely large phylogenies, and the approach could be used to track the evolution of parameters (e.g. R_0) in different regions (sub-trees) of a global tree, as a function of dates (as in Bayesian skyline models), geographical areas, viral variants etc. This new decomposition approach and the corresponding algorithm, named 'subtree picker', have been added to PhyloDeep and described in Methods (page 7).

2. Guidance on which representation to use?

Related to above, I'd like to see a more thorough investigation of the differences between the two representations? Can you find scenarios where one representation would be preferable over the other? Does one take longer to train? Is there a difference in generalizability?

Thank you for this point, we added the following subsection, addressing all these issues (see also above comments and changes regarding generalizability):

SS is simpler, but CBLV has high potential for application to new models

FFNN-SS and CNN-CBLV show similar accuracy across all settings (Fig. 3, Supplementary Tab. 1-2), including when predicting huge trees from their subtrees (Fig. 4). The only exception is the prediction of large trees using NNs trained with small trees (Supplementary Fig. 4), where FFNN-SS is superior to CNN-CBLV, but this goes beyond the recommended use of the approach, as only a part of the (large) query tree is given to the (small) CNN-CBLV.

However, the use of the two representations is clearly different, and it is likely that with new models and scenarios their accuracy will differ. SS requires a simpler architecture (FFNN) and is trained faster (e.g., 5 hours with large BDSS trees), with less training instances (Supplementary Fig. 6). However, this simplicity is obtained at cost of a long preliminary work to design appropriate summary statistics for each new model, as was confirmed in our analyses of BDSS simulations. To estimate the parameters of this model, we added summary statistics on

transmission chains on top of the SS taken from Saulnier et al. [19]. This improved the accuracy of superspreading fraction estimates of the FFNN-SS, so that it was comparable to the CNN-CBLV, while the accuracy for the other parameters remained similar (Supplementary Fig. 7). The advantage of the CBLV is its generality, meaning there is no loss of information between the tree and its representation in CBLV regardless of which model the tree was generated under. However, CBLV requires more complex architectures (CNN), more computing time in the learning phase (150 hours with large BDSS trees) and more training instances (Supplementary Fig. 6). Such an outcome is expected. With raw CBLV representation, the convolutional architecture is used to “discover” relevant summary statistics (or features, in machine learning terminology), which has a computational cost.

In fact, the two representations should not be opposed. An interesting direction for further research would be to combine them (e.g., during the FFNN phase), to possibly obtain even better results. Moreover, SS are still informative and useful (and quickly computed), in particular to perform sanity checks, both a priori and a posteriori (Fig. 5, Supplementary Fig. 8), or to quickly evaluate the predictability of new models and scenarios.

3. What is the impact of the additional features added to the SS representation.

The authors introduced new summary statistics. I'd like to know more about the performance with and without these additional features.

In the revised version, the accuracy of all parameter estimates for BDSS is provided in Supplementary Fig. 7, with and without these additional features, and compared to CBLV. To summarize: the accuracy for the superspreading fraction (the most difficult parameter) is substantially improved with the new features and becomes similar to CNN-CBLV's, while the accuracy for the other parameters remains similar. Note, moreover, that the results for BD and BDEI were obtained with SS including these new features. All this is provided and explained in the revised version (see new subsection above and Supplementary Fig. 7).

Also, how well does a joint representation perform, where you combine the two current representations (based on summary statistics, and based on tree topology/branch lengths).

Combining both representations is certainly an interesting direction for further research. However, this imposes more complex NN architectures; for example, to incorporate the SS in the FFNN phase, after CNN and feature extraction from CBLV. Note, moreover, that the predictions of both approaches are highly correlated (close to 1 for most parameters of the three models), meaning that there is likely little room for improvement. Thus, we decided to leave this research direction for future works, and to give some indications in the Discussion (page 15).

4. Compute likelihood of solutions identified by PhyloDeep

To better understand where the improvement of performance relative to BEAST2 comes from, it would be good to evaluate the likelihood of the solutions identified by PhyloDeep. Do they have larger likelihood than BEAST2 solutions?

We fully understand this demand, but a problem is that computing the likelihood is generally difficult (if not impossible) in phylodynamics, hence the numerous ABC and likelihood-free methods.

However, with the simplest birth-death (BD) model we have a closed form solution to compute the likelihood function, and we applied Referee's suggestion to our 'large' dataset, where BEAST2 and our NNs have similar accuracy (Fig. 3). We also computed the likelihood for the 'true' parameter values used to simulate the trees, in order to have an independent and solid assessment of the performance of the various methods. If a given method tends to produce higher likelihood than the one obtained with the true parameters values, then it performs "well enough" in terms of likelihood optimization, as optimizing further should not result in higher accuracy. The results (Supplementary Tab. 3) were as follows: (i) all methods (BEAST2, FFNN-SS and CNN-CBLV) obtained higher likelihood values than those obtained with true parameter values for ~70% of the trees, with a significant average difference; (ii) the difference of likelihood values between BEAST2, FFNN-SS and CNN-CBLV was non-significant, which explains their similar accuracy. These results are remarkable, as the NNs do not explicitly optimize the likelihood function associated with the model but use a radically different simulation-based learning approach.

Applying the same to BDEI and BDSS turned out to be impossible, as we do not have closed form solutions, and BEAST2 does not converge for several datasets due to numerical issues in likelihood computation and possible local optima (Fig. 3). Using BEAST2, we were unable to compute the likelihood value of our estimates and the one of the true parameter values, for a large fraction of trees. However, for the partial results we obtained (not shown), the figure seems to be similar to that with BD: the NNs obtain highly likely solutions, with similar likelihood as BEAST2's (when it converges and produces reasonable estimates), and significantly higher likelihood than that of the true parameter values.

All this is explained and detailed in the manuscript (pages 8-9), Methods (page 18) and Supplementary Tab. 3.

5. Additional real dataset. Preferably a large-scale one.

Finally, I think the impact of the method/paper can be largely increased if you would consider an additional large-scale dataset.

We agree with the referee on the importance of analysing large data sets and trees, as they are becoming increasingly common today. However, hardly any existing method can accurately

estimate phylodynamics models with trees having (say) >1,000 tips (see our difficulties with BEAST2 and trees with <500 tips). When we submitted the first version of the paper, we were not sure how our NNs could be applied to very large trees, especially with CBLV (with SS it is still possible to summarize big trees using a few dozens of well-chosen features, but with the possible risk of losing essential information). In the revised version, we proposed, implemented and evaluated a solution based on disjoint subtrees extraction, estimation and averaging (see above). To assess this novel approach, we decided to use 'huge' simulated trees (5,000 to 10,000 tips) rather than a real tree, where the actual value of the parameters is often questionable and subject to debate. Results (pages 7, 9-10, Fig. 4) are quite convincing, with remarkably fast and accurate inference (see above), meaning that this approach open the way for new applications of phylodynamics, which were just impossible before. We thank the referee for his/her suggestion, which prompted us to further developments and clearly improved the paper in our opinion.

Minor comments:

* More emphasis on PhyloDeep

The first mention of PhyloDeep occurs very late in the paper, almost like an afterthought. I think it should be featured more prominently, e.g. in the Abstract and Introduction, especially if you want this to be a tool to be used by the community.

Done, in both Abstract and Introduction, we thank the referee for his/her suggestion.

* Line 452: maximal => maximum or largest

Done.

Reviewer #2 (Remarks to the Author):

Review

Deep learning from phylogenies to uncover the transmission dynamics of epidemics

By Voznica et al.

The authors propose an original likelihood-free, simulation-based approach grounded on deep learning to infer the dynamics of epidemics from genetic data. They compare different versions of their approach to existing approaches currently used in the domain. They apply the method to an existing real data set dealing with the HIV epidemic in Zurich, which was already analysed and leads the authors to refine the knowledge about the determinants of HIV transmission in Zurich. The short discussion gives several perspectives for extending the application domain of

their approach. Codes implementing the approach are provided within the Python package PhyloDeep (note that I have not tested the codes).

This study is particularly well designed and many aspects are explored (most of the interrogations that I had during the reading of the main text are actually treated in the supplementary material). The main text is clear and, as I write above, generally adequately complemented by supporting information. The methods are well described and relevant.

The following concerns more specific points.

The title, DEEP LEARNING FROM PHYLOGENIES TO UNCOVER THE TRANSMISSION DYNAMICS OF EPIDEMICS, induces a confusion since the inference of the transmission dynamics may refer to the estimation of 'who infected whom', whereas the authors' objective is upstream: selecting a transmission model and estimating its parameters. Hence, I wonder if a title like DEEP LEARNING FROM PHYLOGENIES TO INFER THE DETERMINANTS OF DISEASE TRANSMISSION DYNAMICS (or something approaching) would be more adequate.

Thank you for this point, we agree that "transmission" can be confusing and changed the title to:

DEEP LEARNING FROM PHYLOGENIES TO INFER THE EPIDEMIOLOGICAL DYNAMICS OF OUTBREAKS

To facilitate the reading across this rich piece of work, supplementary figures and tables should be ordered as they appear in the main text.

Done.

After reading the introduction, I was unsure whether you were embedding your method in the framework of ABC or not. If I understood correctly, you do not, and you simply mention ABC, and more specifically the paper by Saulnier et al., because you recycle summary statistics that were proposed by Saulnier and her colleagues. At l.68, the term 'rejection-free' puts the reader on the track that you do not develop an ABC approach, but you should make it more explicit at the transition from the paragraph about the Saulnier's paper (l.60-67) to the next paragraph (l.68-80).

This has been clarified. In fact, our approach is a continuation of regression-based ABC. We wrote (blue part is new; page 4):

...To address this issue Saulnier et al. [19] developed a large set of summary statistics. In addition, they used a regression step to select the most relevant statistics and to correct for the discrepancy between the simulations retained in the rejection step and the analyzed phylogeny. They observed that the sensitivity to the rejection parameters were greatly attenuated thanks to regression (see also Blum et al. [20]).

Our work is a continuation of regression-based ABC, and aims at overcoming its main limitations. Using the approximation power of currently available neural network architectures, we propose a likelihood-free method relying on deep learning from millions of trees of varying size simulated within a broad range of parameter values. By doing so, we bypass the rejection step, which is both time consuming with large simulation sets, and sensitive to the choice of the distance function and summary statistics...

I.122: The authors should more exactly specify here what they call the 'sampling probability' and what means 'known'. This is clear later in the paper, but is ambiguous at this stage of the paper.

This has been clarified (page 6).

I.196-197: The convergence issue only concerns BEAST2, doesn't it?

Yes, this has been clarified (page 8).

In the discussion, the authors should evoke the question about how their approaches scale up with larger data sets than those considered in the application and simulations, with larger trees, lower sampling probability, and models with more parameters. In particular, is there a need for a much larger number of simulations (than 4M) for training the deep learning tools that the authors proposed to use in these cases? Supp. Fig. 3 and 4 partly tackle this question and, if my interpretation is correct, SF3 states that the performance is relatively stable in terms of model selection when the tree size increases, and SF4 states that the performance in terms of parameter estimation accuracy increases with the number of simulations. The authors could make a synthesis of this type of results in the discussion and extrapolate (to some extents) for answering the other dimensions of the above-mentioned question.

Thank you for raising these points, which are addressed here and there, and summarized in the Discussion, where we added (pages 15-16):

A key issue in both phylodynamics and machine learning applications is scalability. Our results show that very large phylogenies can be analysed very efficiently (~1 minute for 10,000 tips), with resulting estimates more accurate than with smaller trees (Fig. 4), as predicted by learning theory. Again, as expected, more complex models require more training instances, especially BDSS using CBLV (Supplementary Fig. 3), but the ratio remains reasonable, and it is likely that complex (but identifiable) models will be handled efficiently with manageable training sets. Surprisingly, we did not observe a substantial drop of the accuracy with lower sampling probabilities (results not shown). To analyse very large trees, we used a decomposition into smaller, disjoint subtrees. In fact, all our NNs were trained with trees of moderate size (<500 tips). Another approach would be to learn directly from large trees. This is an interesting direction for further research, but this poses several difficulties. The first is that we need to simulate these very large trees, and a large number of them (millions or more). Then, SS is the easiest representation to learn, but at the risk of losing essential information, which means that new summary statistics will likely be needed for sufficiently complete representation of very

large phylogenies. Similarly, with CBLV more complex NN architectures (e.g., with additional and larger kernels in the convolutional layers) will likely be needed, imposing larger training sets. Combining both representations (e.g., during the FFNN phase) is certainly an interesting direction for further research. Note, however, that the predictions of both approaches for the three models we studied are highly correlated (Pearson coefficient nearly equal to 1 for most parameters), which means that there is likely little room for improvement (at least with these models).

Fig. 4 and Supp. Fig. 8: the authors interestingly show that observed summary statistics for HIV are within the `simulated envelope' of summary statistics throughout an analysis of the first two axes of a PCA. It would be interesting as well to perform the a priori check for the row summary statistics (without PCA). Since there are many summary statistics, the authors could provide a relatively concise table indicating, for each SS (i.e., marginally), to which quantile the observed value corresponds. This table could also be summarized into an histogram providing the distribution of the afore-mentioned quantiles. The table and the histogram would more precisely indicate how the class of used models represent real data.

Thanks for the suggestion; we have added this functionality to PhyloDeep (page 19 in Methods). The SSs of the input tree are provided to the user, along with the corresponding [min, max] values in our simulations. With the HIV dataset, some SSs rejected the BD and BDEI models, which consistently have probability 0 in model selection (Supplementary Tab. 5 and Fig. 8).

Supp. Table 1 should include the information given at l.823-825 that a different prior is used for the infectious period in the numerical experiment and in the application. Looking at Fig. 4, it seems to me that a different prior is also used for $X_{\{SS\}}$, but I am maybe wrong and I maybe missed this information in the text.

In Fig. 5 (previously Fig. 4) we display the posterior distributions of the parameters, not the priors which were the same as in the simulations (in fact, our "priors" correspond to the simulation parameters, displayed in Supplementary Table 1). This has been clarified in the figure legend.

**DEEP LEARNING FROM PHYLOGENIES TO UNCOVER**
**THE EPIDEMIOLOGICAL DYNAMICS OF OUTBREAKS**

**AUTHORS**

Voznica J^{1,2,3*}, Zhukova A^{1,4,5,6*}, Boskova V⁷, Saulnier E¹, Lemoine F^{1,4}, Moslonka-Lefebvre M¹, Gascuel O^{1,8*}

**AFFILIATIONS**

¹ Institut Pasteur, Université Paris Cité, Unité Bioinformatique Evolutive, Paris, FRANCE

² Université de Paris, Paris, FRANCE

³ Institut de Biologie de l'École Normale Supérieure, Ecole Normale Supérieure, CNRS, INSERM, Université Paris
Sciences et Lettres, Paris, FRANCE

⁴ Institut Pasteur, Université Paris Cité, Bioinformatics and Biostatistics Hub, Paris, FRANCE

⁵ Institut Pasteur, Université Paris Cité, Epidemiology and Modelling of Antibiotic Evasion, Paris, FRANCE

⁶ Université Paris-Saclay, UVSQ, Inserm, CESP, Villejuif, FRANCE

⁷ Center for Integrative Bioinformatics Vienna, Max Perutz Labs, University of Vienna and Medical University of
Vienna, Vienna, AUSTRIA

⁸ Institut de Systématique, Evolution, Biodiversité (UMR 7205 - CNRS, Muséum National d'Histoire Naturelle, SU,
EPHE, UA), Paris, FRANCE

*** CO-CORRESPONDING AUTHORS**

jakub.voznica@pasteur.fr (JV), anna.zhukova@pasteur.fr (AZ), olivier.gascuel@mnhn.fr (OG)

**ABSTRACT**

[revised manuscript text omitted]

dynamic inference in BEAST 2 (2020). Preprint at [https://www.biorxiv.org/con-](https://www.biorxiv.org/content/10.1101/2020.01.06.895532v1.full.pdf)
[tent/10.1101/2020.01.06.895532v1.full.pdf](https://www.biorxiv.org/content/10.1101/2020.01.06.895532v1.full.pdf)
- 13. Bouckaert, R. *et al.* BEAST 2: A Software Platform for Bayesian Evolutionary Analysis. *PLoS Computa-*
*tional Biology* **10(4)**, e1003537 (2014).
- 14. Boskova, V., Stadler, T., Magnus, C. The influence of phylodynamic model specifications on parameter
estimates of the Zika virus epidemic. *Virus Evolution* **4(1)**, vex044 (2018).
- 15. Vaughan, T.G., Sciré, J., Nadeau, S.A., Stadler, T. Estimates of outbreak-specific SARS-CoV-2 epidemio-
logical parameters from genomic data (2020). Preprint at [https://www.medrxiv.org/con-](https://www.medrxiv.org/content/10.1101/2020.09.12.20193284v1.full.pdf)
[tent/10.1101/2020.09.12.20193284v1.full.pdf](https://www.medrxiv.org/content/10.1101/2020.09.12.20193284v1.full.pdf)
- 16. Rubin, D.B. Bayesianly Justifiable and Relevant Frequency Calculations for the Applies Statistician. *The*
*Annals of Statistics* **12**, 1151-72 (1984).
- 17. Beaumont, M.A., Zhang, W., Balding, D.J. Approximate Bayesian Computation in Population Genetics.
*Genetics* **164**, 2025-35 (2002).
- 18. Csilléry, K., Blum, M.G.B., Gaggiotti, O.E., François, O. Approximate Bayesian Computation (ABC) in
practice. *Trends in Ecology & Evolution* **25**, 410-8 (2010).
- 19. Saulnier, E., Gascuel, O., Alizon, S. Inferring epidemiological parameters from phylogenies using regres-
sion-ABC: A comparative study. *PLoS Comp. Biol.* **13(3)**, e1005416 (2017).
- 20. Blum, M.G.B. *Handbook Of Approximate Bayesian Computation Ch. Regression approaches for ABC.* 71–
85. (Chapman and Hall/CRC Press, Boca Raton, 2018).
- 21. LeCun, Y., Kavukcuoglu, K., Farabet, F. Convolutional networks and applications in vision. *Proc. IEEE*
*Int. Symp. Circuits Syst.* 253-6 (2010).
- 22. Krizhevsky, K., Sutskever, I., Hinton, G.E. ImageNet Classification with Deep Convolutional Neural Net-
works. *Advances in neural information processing systems* 1097-105 (2012).
- 23. Chattopadhyay, A., Hassanzadeh, P., Pasha, S. Predicting clustered weather patterns: A test case for appli-
cations of convolutional neural networks to spatio-temporal climate data. *Sci. Rep.* **10**, 1317 (2020)
- 24. The Swiss HIV Cohort Study *et al.* Cohort Profile: The Swiss HIV Cohort Study. *International Journal of*
*Epidemiology* **39**, 1179–89 (2010).

- 25. Rasmussen, D.A., Kouyos, R., Günthard, H.F., Stadler, T. Phylodynamics on local sexual contact networks.
*PLOS Comp. Biol.* **13(3)**, e1005448 (2017).
- 26. Colijn, C. & Plazzotta, G. A metric on phylogenetic tree shapes. *Systematic Biology* **67**, 113–26 (2018).
- 27. Liu, P., Gould, M., Colijn, C. Analyzing Phylogenetic Trees with a Tree Lattice Coordinate System and a
Graph Polynomial, *Systematic Biology*, in press (2022). Preprint at <https://doi.org/10.1093/sysbio/syac008>
- 28. Lewitus, E. & Morlon, H. Characterizing and Comparing Phylogenies from their Laplacian Spectrum. *Sys-*
*tematic Biology* **65**, 495-507 (2016).
- 29. Kim, J., Rosenberg, N.A., Palacios, J.A. Distance metrics for ranked evolutionary trees. *Proceedings of the*
*National Academy of Sciences* **117**, 28876-86 (2020).
- 30. Cormen, T.H., Leiserson, C.E., Rivest, R.L., Stein, C. *Introduction To Algorithms*. 286-307 (The MIT
Press, Cambridge, 2009).
- 31. Bengio, Y. *Neural Networks: Tricks Of The Trade*, Ch. *Practical Recommendations for Gradient-Based*
*Training of Deep Architectures*. (Springer, Berlin, Heidelberg 2002).
- 32. Gelman, A., Carlin, J.B., Stern, H.S., Rubin, D.B. *Bayesian Data Analysis: Second Edition*. (Chapman and
Hall/CRC Press, Boca Raton, 2004).
- 33. Baele, G. *et al.* Improving the accuracy of demographic and molecular clock model comparison while ac-
commodating phylogenetic uncertainty. *Mol. Biol. Evol.* **29**, 2157-67 (2012).
- 34. Kouyos, R.D. *et al.* Molecular epidemiology reveals long-term changes in HIV type 1 subtype B transmis-
sion in Switzerland. *J. Infect. Dis.* **201**, 1488-97 (2010).
- 35. May, R.M. & Anderson, R.M. Transmission dynamics of HIV infection. *Nature* **326**, 137–142 (1987).
- 36. Brenner, B.G. *et al.* Quebec Primary HIV Infection Study Group. High rates of forward transmission events
after acute/early HIV-1 infection. *J. Infect. Dis.* **195**, 951-9 (2007).
- 37. Gueler, A. *et al.* Swiss National Cohort Life expectancy in HIV-positive persons in Switzerland. *AIDS* **31**,
427-436 (2017).
- 38. Rasmussen, D.A., Volz, E.M., Koelle, K. Phylodynamic Inference for Structured Epidemiological Models.
*PLoS Comput. Biol.* **10(4)**, e1003570 (2014).
- 39. Volz, E.M. & Siveroni, I. Bayesian phylodynamic inference with complex models. *PLoS Comput. Biol.*
**14(11)**, e1006546 (2018).

- 40. MacPherson, A., Louca, S., McLaughlin, A., Joy, J.B., Pennell, M.W. Unifying Phylogenetic Birth–Death
Models in Epidemiology and Macroevolution, *Systematic Biology*, **71(1)**, 172–189 (2022).
- 41. Sanchez, T., Cury, J., Charpiat, G., Jay, F. Deep learning for population size history inference: Design,
comparison and combination with approximate Bayesian computation. *Mol. Ecol. Resour.* **00**, 1-16. (2020).
- 42. Dunn, D. & Pillay, D. UK HIV drug resistance database: background and recent outputs. *J. HIV Ther.* **12**,
97–8 (2007).
- 43. Shu, Y. & McCauley, J. GISAID: Global initiative on sharing all influenza data - from vision to reality.
*Euro Surveill.* **22**, 30494 (2017).
- 44. Minh, B.Q. *et al.* IQ-TREE 2: New models and efficient methods for phylogenetic inference in the genomic
era. *Mol. Biol. Evol.* **37**, 1530–1534 (2020).
- 45. Kozlov, A.M., Darriba, D., Flouri, T., Morel, B., Stamatakis, A. RAxML-NG: a fast, scalable and user-
friendly tool for maximum likelihood phylogenetic inference. *Bioinformatics* **35**, 4453–4455 (2019).
- 46. Guindon, S. *et al.* New Algorithms and Methods to Estimate Maximum-Likelihood Phylogenies: Assessing
the Performance of PhyML 3.0. *Systematic Biology* **59**, 307-21 (2010).
- 47. Sagulenko, P., Puller, V., Neher, R.A. TreeTime: Maximum-likelihood phylodynamic analysis. *Virus*
*Evol.* **4(1)**, vex042 (2018).
- 48. To, T.H., Jung, M., Lycett, S., Gascuel, O. Fast Dating Using Least-Squares Criteria and Algorithms. *Syst*
*Biol.* **65**, 82-97 (2016).
- 49. Volz, E.M. & Frost, S.D.W. Scalable relaxed clock phylogenetic dating. *Virus Evol.* **3(3)**, vex025 (2017).

Reviewers' Comments:

Reviewer #1:

Remarks to the Author:

I thank the authors for their detailed response to my previous comments, which have been satisfactorily addressed in the revised manuscript. This work will be of interest to the readership of this journal.